# TYPE-II SADDLES AND PROBABILISTIC STABILITY OF STOCHASTIC GRADIENT DESCENT

## ABSTRACT

Characterizing and understanding the dynamics of stochastic gradient descent (SGD) around saddle points remains an open problem in neural network optimization. We identify two distinct types of saddle points, demonstrating that Type-II saddles pose a significant challenge due to vanishing gradient noise, which makes them particularly difficult for SGD to escape. We show that the dynamics around these saddles can be effectively modeled by a random matrix product process, allowing us to apply concepts from probabilistic stability and Lyapunov exponents. By leveraging ergodic theory, we establish that saddle points can be either attractive or repulsive for SGD, leading to a classification of four distinct dynamic phases based on the gradient's signal-to-noise ratio near the saddle. We apply the theory to the training at the initial stage of neural networks, explaining an intriguing phenomenon that neural networks are prone to be stuck at the initialization point at a larger learning rate. Our results offer a novel theoretical framework for understanding the intricate behavior of SGD around saddle points, with implications for improving optimization strategies in deep learning.

## 1 INTRODUCTION

Understanding the optimization landscape is fundamental to unlocking the full potential of deep learning. Despite the remarkable successes of neural networks, their training often involves navigating complex loss surfaces riddled with saddle points, which significantly influence the behavior and efficiency of stochastic gradient descent (SGD). Recent research suggests that the trajectory of SGD predominantly moves from one saddle point to another during training (Jacot et al., 2021; Abbe et al., 2023), making the ability to characterize and understand these dynamics an urgent priority for the deep learning community.

In this work, we delve into this critical problem by introducing a classification of saddle points into two distinct types: (1) Type-I, where gradient noise persists at the saddle, and (2) Type-II, where the gradient noise vanishes entirely (Def. 1). Our primary focus is on the Type-II saddles, which pose a greater challenge due to their tendency to trap SGD, even in the presence of a negative eigenvalue in the Hessian matrix.

Our main contributions are:

1. **Classification of saddles**: we propose to classify saddle points in neural networks into two types, Type-I where the noises of the gradient do not vanish in the escape directions, and Type-II saddles where the gradient noises vanish in the escape directions;
2. **Probabilistic Stability**: we propose using the probabilistic stability and Lyapunov exponents to study the attractivity of Type-II saddle points of neural networks, bridging the conventional study of dynamical systems in control theory and ergodic theory to study the linear dynamics of SGD close stationary points;
3. **Phases and Difficulty of Network Initialization**: we use the probabilistic stability to show that close to a Type-II saddle point, SGD has at least four different phases of learning, which we apply to understand the difficulty of using a large learning rate during the initialization of neural nets.

This work is organized as follows. Section 2 discusses the most relevant works. Section 3 discusses the difference between the two types of saddle points encountered in training neural networks. Section 4 studies the probabilistic stability and Lyapunov exponent of SGD around these fixed points.

Section 5 applies the theory to relevant problems in deep learning, especially the initialization of neural networks. All proofs and experimental details are presented in the Appendix.

## 2 RELATED WORKS

**Dynamical stability**. Dynamical stability centers around studying the attractivity and repulsiveness of fixed points. From an optimization perspective, stability can be seen as a worst-case guarantee for optimization because the training algorithm cannot converge to a fixed point that is unstable. Conceptually, the work closest to ours is Wu et al. (2018), which uses the variance of SGD to characterize the set of solutions that are preferred by SGD, namely the solutions that are flat and have a rather weak noise. Wang & Jacot (2023) extends the discussion to explore the rank of solutions at local minima. However, as our result shows, moment-based notions of stability cannot be applied to saddle points. Instead, we propose to study this problem with the tools of probabilistic stability, which has been studied extensively in control theory (Khas' minskii, 1967; Eckmann & Ruelle, 1985; Teel et al., 2014). The connection between Lyapunov exponents and probabilistic stability for the random matrix product problem is well known (Diaconis & Freedman, 1999), but its relevance for studying saddle points in deep learning is unclear. Prior works have used similar notions to study the convergence to a local minimum (Gurbuzbalaban et al., 2021; Hodgkinson & Mahoney, 2021). In contrast, our focus is on the attractivity of saddle points. Recently, Chen et al. (2024) introduced the concept of stochasitc collapse, which can be seen as a manifestation of the probabilistic stability when the dynamics is in continuous time.

**Saddle points and Initialization**. Due to the nonlinear nature of the loss function of neural networks, escaping from saddle points has been an important problem in deep learning (Dauphin et al., 2014; Ge et al., 2015; Mertikopoulos et al., 2020; Vlaski & Sayed, 2022; Pemantle, 1990). As a primary example, Dauphin et al. (2014) mainly focuses on studying saddles from the perspective of gradient descent, where there is no difference between Type-I and Type-II saddles. See the next section for more discussion about prior literature on saddles. A long puzzling observation is that using a large learning rate at initialization often causes the network to be stuck at initialization for a long time (Loshchilov & Hutter, 2016), and warmup are often required to successfully initiate training. Our result shows that this problem is due to having large learning rate close to a Type-II saddle.

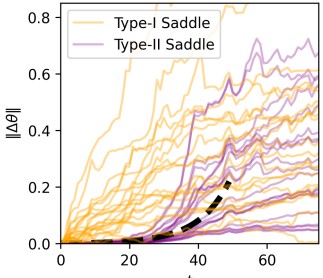

Figure 1: Escaping from two types of saddles in a ReLU network under SGD. We see that for the escapes from Type-I saddles, the escaping process starts immediately for every trajectory. For Type-II saddles, the escape only starts significantly after the training starts, despite the gradient noise. The black dashed line shows an exponential fit to the Type-II, implying the connection of the problem to Lyapunov exponents. See Appendix B.1 for details on the construction of these saddles.

## 3 TWO TYPES OF SADDLE POINTS

Escaping from saddle points is a well-known and fundamental problem in the optimization of neural networks. Conventional wisdom believes that as long as the saddle points are strict (having at least one negative eigenvalue in the Hessian), escaping from it is easy (Sun et al., 2020). This holds true for gradient descent training. However, as we will show, strictness is insufficient to guarantee an escape when the training proceeds with SGD.

Let $\ell'(\theta, x')$ be loss for a data point $x'$ and parameters $\theta$. The per-batch loss is defined as the empirical average of $\ell'$ over a minibatch $B$ of an arbitrary batch size $S$:

$$\ell(\theta, B) = \frac{1}{S} \sum_{x' \in B} \ell'(\theta, x'). \tag{1}$$

Because $B$ can be seen as a random vector, we denote $B$ as $x$ to write the per batch loss as $\ell(\theta, x)$. At every step, the minibatch is independently sampled to perform the SGD update:

$$\theta_{t+1} = \theta_t - \lambda \nabla_\theta \ell(\theta_t, x), \tag{2}$$

where $\lambda$ is the learning rate, and $\theta_t$ is the parameter at the $t$-th iteration of SGD.

We propose to classify the saddle points into two types and show that the Type-II saddles are much more difficult to escape than the Type-I saddles. A point $\theta$ is said to be a saddle point if $L(\theta) :=$

$\mathbb{E}[\nabla_\theta \ell] = 0$, where $\mathbb{E} := \mathbb{E}_{x \sim p(x)}$ denotes expectation of the data $x$ with respect to the distribution of training data $p(x)$, and $\theta$ is not a local minimum. This implies that

$$\|\mathbb{E}[\theta_{t+1} - \theta_t]\| = O(\Delta\theta_t), \tag{3}$$

where $\Delta\theta_t := \|\theta_t - \theta^*\|$ is the Euclidean distance from the saddle point $\theta^*$. However, when stochasticity is taken into account, there are two types of saddles that satisfy this condition:

$$\theta_{t+1} - \theta_t = \lambda r_t + O(\Delta\theta_t) \quad \text{(Type-I saddle);} \tag{4}$$

$$\theta_{t+1} - \theta_t = \lambda \hat{H}(x_t)\Delta\theta_t + O(\Delta\theta_t^2) \quad \text{(Type-II saddle),} \tag{5}$$

where $\lambda$ is the learning rate, $r_t$ is a random vector with a non-vanishing variance that is independent of $\theta$, and $\hat{H}(x_t) = \nabla_\theta^2 \ell(\theta^*, x_t)$ is a random symmetric matrix that is a function of the random data points $x_t$.

The source of the randomness is due to the minibatch sampling, where $\hat{H}$ is identified as the batch-wise Hessian matrix. There is a key difference between the two types of saddles: Type-I saddles have a nonvanishing noise at the saddle, and this noise makes it possible for SGD to escape from the saddle. Meanwhile, the Type-II saddles have a vanishing noise at the saddle, hampering an efficient escape. See Figure 1 for a comparison of escaping from the two types of saddles during neural network training. Here, the Type-II saddle is induced by the permutation symmetry between neurons in the hidden layer. This discussion motivates the following definition for Type-II saddles. Let $H(\theta) = \nabla_\theta^2 \mathbb{E}[\ell(\theta, x)]$ denote the Hessian of the loss at $\theta$, and $P_\theta$ denote the orthogonal projection to the subspaces where $H(\theta)$ has non-positive eigenvalues. Essentially, $P_\theta$ denotes the escaping directions from the saddle.[1]

**Definition 1.** *Let $\theta$ be a saddle point for $L(\theta)$ with the associated escape projection $P_\theta$. $\theta$ is a Type-II saddle of $\ell(\theta, x)$ if $P_\theta \nabla_\theta \ell(\theta, x) = 0$ for all $x$. $\theta$ is a Type-I saddle if it is not Type-II.*

Since $P_\theta$ is always defined with respect the Hessian $H(\theta)$, we omit its subscript $\theta$ and write $P$ for the rest of the manuscript. This definition generalizes Eq. (5) and setting $P = I$ recovers Eq. (5). The following Lemma shows that the dynamics of SGD around saddle points are can be divided into two types, like Eq. (4) and Eq. (5).

**Proposition 1.** *Let $\theta^*$ be a saddle point with the corresponding escape projection $P$. Then, if $\theta^*$ is a Type-I saddle,*

$$P\theta_{t+1} = P\theta_t - \lambda r_t + O(\Delta\theta_t), \tag{6}$$

*where $r_t := P\nabla_\theta \ell(\theta^*, x_t)$. If $\theta^*$ is a Type-II saddle,*

$$P\theta_{t+1} = P\theta_t - \lambda P^T \hat{H}(x_t) P(\theta_t - \theta^*) + O(\Delta\theta_t^2). \tag{7}$$

*Proof.* Close to a saddle point, the loss per batch is given by the Taylor series:

$$\ell(\theta_t, x_t) = \ell(\theta^*, x_t) + \nabla_\theta \ell(\theta^*, x_t)^T(\theta_t - \theta^*) + \frac{1}{2}(\theta_t - \theta^*)^T \nabla_\theta^2 \ell(\theta^*, x_t)(\theta_t - \theta^*) + O(\Delta\theta_t^2).$$

Inserting the Taylor series to Eq. (2) and applying projection operator $P$, we obtain

$$P\theta_{t+1} - P\theta_t = -\lambda(P\nabla_\theta \ell(\theta^*, x_t) + P^T \hat{H}(x_t) P(\theta_t - \theta^*)) + O(\Delta\theta_t). \tag{8}$$

If $\theta^*$ is Type-II, we have $P\nabla_\theta \ell(\theta^*, x_t) = 0$, and Eq. (8) reduces to Eq. (7).

Otherwise, there exists $x$ such that $P\nabla_\theta \ell(\theta^*, x) \neq 0$. Then, we can define $r_t := P\nabla_\theta \ell(\theta^*, x_t)$ as a non-zero random vector, and Eq. (8) reduces to Eq. (6). This finishes the proof. □

Prior works suggest that Type-II saddle points exist abundantly in the loss function of neural networks and the matrix $P$ depends on the architecture of the network. The main cause of the saddle points in neural networks is the vast number of parameter symmetries, such as permutation symmetry (Fukumizu & Amari, 2000; Simsek et al., 2021; Entezari et al., 2021; Hou et al., 2019), rescaling symmetry (Dinh et al., 2017; Neyshabur et al., 2014), and rotation symmetry (Ziyin et al., 2023).

---

[1]Namely, let $V = \{v_i\}_i$ be the subset of unit eigenvectors of $H(\theta)$ with a nonpositive eigenvalue, then $P = \sum_{v \in V} vv^T$.

Ziyin (2023) proves that close to these symmetry-induced saddle points, the SGD dynamics is always described by the type of dynamics in Eq. (7). Convergence to these symmetry-induced saddles has been identified as a main cause of collapsing of models into low-capacity solutions (Ziyin, 2023). Literature regarding SGD escaping saddle points (e.g., Ge et al. (2015); Mertikopoulos et al. (2020); Vlaski & Sayed (2022); Pemantle (1990)) often takes advantage of the assumption of non-vanishing noise thus studies escaping Type-I saddle. Type-II saddles are thus poorly understood despite their relevance in deep learning.[2]

We will show in the following sections that it is essentially the distribution of $\hat{H}$ that determines the dynamics of SGD close to these saddle points. When we have the same data distribution but a different batch size $S$, the distribution of $\hat{H}$ is different and scales like $1/S$. Also, it is worth commenting that while our focus is on the saddles, our theory also applies to interpolation minima in neural networks. Here, because the loss function for every data point vanishes, and the dynamics around it also obeys an identical form to Eq. (5). The dynamics we consider is more general than that around an interpolation minimum because the Hessians $\hat{H}$ in Eq. (7) are allowed to have both positive and negative eigenvalues, whereas an interpolation minimum only allows for positive semidefinite $\hat{H}$. Therefore, the general solution of Eq. (7) also helps us understand this type of minima better once we restrict the study to PSD Hessians.

## 4 PROBABILISTIC STABILITY OF SGD

In this section, we introduce the main theoretical framework. We first define the probabilistic stability, and then apply it to Type-II saddles.

### 4.1 PROBABILISTIC STABILITY

We use $\|\cdot\|_p$ to denote the $p-$norm of a matrix or vector and $\|\cdot\|$ to denote the case where $p = 2$. The notation $\to_p$ indicates convergence in probability.

**Definition 2.** *A sequence (of random variables) $\{\theta_t\}_t^\infty$ generated by the SGD algorithm is probabilistically stable at a constant vector $\theta^*$ if $\theta_t \to_p \theta^*$.*

A sequence $\theta_t$ converges in probability to $\theta^*$ if $\lim_{t\to\infty} \mathbb{P}(\|\theta_t - \theta^*\| > \epsilon) = 0$ for any $\epsilon > 0$. This is a common tool in control theory to study stability. We will see that this notion of stability is especially suitable for studying the stability of the saddle points in SGD. For deep learning, the sequence of $\theta_t$ is the model parameters obtained by the iterations of the SGD algorithm for a neural network. It is usually impossible to analyze the convergence behavior of the dynamics starting from an arbitrary initial condition. Therefore, we have to restrict to the neighborhood of a stationary point and consider the leading-order SGD dynamics around it.

### 4.2 1D DYNAMICS IS EXACTLY SOLVABLE

An analytically solvable case is when the dynamics lie in a 1d manifold. Let $\hat{H}(x) = h(x)nn^T$ be rank-1 for a random scalar function $h(x)$, and a fixed unit vector $n$ for all data points $x$. Thus, the dynamics simplifies to a 1d dynamics, where $h(x) \in \mathbb{R}$ is the corresponding eigenvalue of $\hat{H}(x)$:

$$\theta_{t+1} = \theta_t - \lambda h(x)nn^T(\theta_t - \theta^*). \tag{9}$$

**Theorem 1.** *Let $\theta_t$ follow Eq. (9). Then, for any distribution of $h(x)$, $n^T(\theta_t - \theta^*) \to_p 0$ if and only if*

$$\mathbb{E}_x[\log|1 - \lambda h(x)|] < 0. \tag{10}$$

The l.h.s. of the inequality can be identified as the Lyapunov exponent for the process. The condition Eq. (10) is a sharp characterization of when a critical point becomes attractive. It also works with

---

[2]While parameter symmetry is the main cause of the Type-II saddles, it is not the only cause. For example, consider a two-layer one-dimensional network with the GeLU activation $\sigma(x) = x\Phi(x)$, where $\Phi(x)$ is the cumulative distribution function of Gaussian distribution. In this case, the network $f(x) = u\sigma(wx)$. With the MSE loss $\frac{1}{2}(f(x) - y)^2$, the loss gradient for each data point is $(f(x) - y)(\sigma(wx), ux\sigma'(wx))$. It is readily verified that the gradient is always zero at the origin, indicating that the origin is a Type-II saddle (assuming it is not a local minimum). However, there is no symmetry in this loss.

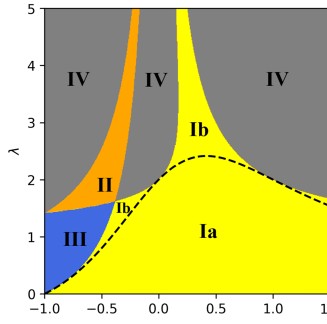 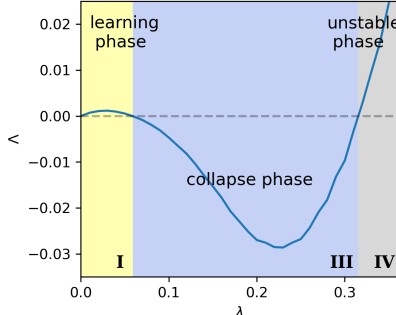

Figure 2: SGD exhibits a complex phase diagram close to a Type-II saddle. **Left**: $a$ denotes the parameter in the data distribution, as discussed in detail in section 5.2. For a matrix factorization saddle point, the dynamics of SGD can be categorized into at least five different phases. Phase **I**, **II**, and **IV** correspond to a successful escape from the saddle. Phase **III** is where the model converges to a low-rank saddle point. Phase **I** corresponds to the case $w_t \to_p u_t$, which signals correct learning. In phase **Ia**, the model also converges in variance. Phase **II** corresponds to stable but incorrect learning, where $w_t \to_p -u_t$. Phase **IV** corresponds to complete instability. **Right**: the phases of SGD can quantified by the sign of the Lyapunov exponent $\Lambda$. Where $\Lambda < 0$, SGD collapses to a saddle point; when $\Lambda > 0$, SGD escapes the saddle and enters an escaping phase. The two escaping phases are qualitatively different. For a small learning rate, the model is in a learning phase due to the repulsiveness of the saddle point at a small learning rate. For a large learning rate, SGD escapes the saddle due to complete loss of stability.

weight decay. When weight decay is present, the diagonal terms of $\hat{H}$ are shifted by $\gamma$, and so $h = h' + \gamma$. At what learning rate is the condition violated? To leading orders in $\lambda$, this can be identified by expanding the logarithmic term up to the second order in $\lambda$:

$$\mathbb{E}_x[\log|1 - \lambda h(x)|] = -\lambda\mathbb{E}[h(x)] - \frac{1}{2}\lambda^2\mathbb{E}_x[h(x)^2] + O(\lambda^3).$$

To leading order, the dynamics follows the sign of $\mathbb{E}[h(x)]$, in agreement with the observation that the GD algorithm always escapes a strict saddle. When the second-order term is significant, the fluctuation of $h(x)$ now decides the attractivity of the critical point, which is attractive if

$$\lambda > 2\frac{-\mathbb{E}[h(x)]}{\mathbb{E}[h(x)^2]}. \tag{11}$$

This condition directly points to the reason why Type-II saddle points are difficult (or impossible) to escape: the critical point can be attractive even if $h < 0$, when the point has become a saddle point. The r.h.s. of the condition also has a natural interpretation as a signal-to-noise ratio (SNR) in the gradient. The numerator is the Hessian of the original loss function, which determines the signal in the gradient. The denominator is the strength of the gradient noise in the minibatch (Wu et al., 2018). An illustration of this solution is given in Figure 2. We show the probabilistic stability conditions for a rank-1 saddle point with a rescaling symmetry (see Section 5.2). The loss function is $\ell(u, w) = -xyuw + o(u^2 + w^2)$. Here, the data points $xy = 1$, and $xy = a$ are sampled with equal probability. These saddles appear naturally in matrix factorization problems and also in recent sparse learning algorithms (Poon & Peyré, 2021; 2022; Ziyin & Wang, 2023; Kolb et al., 2023).

### 4.3 Insufficiency of Norm-Stability

It is very instructive to compare with the convergence in the $L_p$ sense, which is the standard alternative notion of attractivity.

**Definition 3.** *A sequence $\{\theta_t\}_t^\infty$ is $L_p$-norm stable at $\theta^*$ if $\lim_{t\to\infty} \mathbb{E}\|\theta_t - \theta^*\|_p^p \to 0$.*

Theorem 1 provides a perfect example to compare the probabilistic stability with the norm-stability. If SGD converges to a point in $L_p$-norm, it also converges in probability. Thus, norm stability is a more restricted notion than probabilistic stability.

The following result shows that saddle points are always unstable for moment stability, which is constructively established by the following proposition.

**Theorem 2.** *Let $\theta_t$ follow Eq. (7) around a critical point $\theta^*$ of an arbitrary loss. Then, for any fixed $\lambda$,*

1. *there exists a data distribution such that $\theta_t$ is probabilistically stable but not $L_p$-stable;*
2. *if $\theta^*$ is a saddle point and $p \geq 1$, the set of $\theta_0$ that is $L_p$-stable has Lebesgue measure zero.*

Therefore, the $L_p$-stability is not useful in understanding the stability of SGD close to Type-II saddles because the parameters always escape in expectation. One reason is that the outliers strongly influence the $L_p$ norm in the data, whereas the probabilistic stability is robust against such outliers, a point we will discuss in the next section. This theorem points to a special property of Type-II saddles. Namely, that studying Type-II saddles requires a set of special mathematical tools to understand.

### 4.4 LYAPUNOV EXPONENT AND PROBABILISTIC STABILITY

A problem of the probabilistic stability is that it is not a quantitative notion. This problem can be avoided by noticing that the dynamics in Eq. (5) is the same as a random matrix product:

$$\theta_t - \theta^* = (I - \lambda\hat{H}_{t-1})(\theta_{t-1} - \theta^*) = \prod_i^{t-1} Z_i(\theta_0 - \theta^*), \tag{12}$$

where $Z_t = I - \lambda\hat{H}_t$. By the Furstenberg-Kesten theorem (Furstenberg & Kesten, 1960), one can prove that the condition for the probabilistic attractivity of the saddle point under SGD is the same as the Lyapunov exponent of the process being negative. The *maximal Lyapunov exponent* of a point $\theta^*$ is defined as

$$\Lambda = \max_{\theta_0} \lim_{t \to \infty} \frac{1}{t}\mathbb{E}\left[\log\frac{\|\theta_t - \theta^*\|}{\|\theta_0\|}\right]. \tag{13}$$

The expectation is taken over the random samplings of the SGD algorithm. In general, $\Lambda$ does not vanish. The following theorem shows that SGD is probabilistically stable at a point if and only if its Lyapunov exponent is negative. The maximum Lyapunov exponent is initialization-independent.[3]

**Theorem 3.** *Assuming that $\Lambda \neq 0$, the linearized dynamics of SGD is probabilistically stable at $\theta^*$ for any $\theta_0$ if and only if $\Lambda < 0$.*

This theorem means that one can directly quantify the notion of probabilistic stability using the Lyapunov exponent. When applied to a saddle point problem, a negative $\Lambda$ implies that SGD will not be able to escape the saddle. This result can also be proved by adapting the classical results in Bougerol & Picard (1992). Note that for a finite dataset size, one can always choose a learning rate such that $I - \lambda\hat{H}$ is all positive with probability 1, this further makes it possible to apply the central limit theorem of the classical Furstenberg-Kesten theory.

For a finite-size dataset, it is straightforward to find an upper and lower bound for $\Lambda$. we can define $r_{\max}$ to be larger than the absolute value of the eigenvalues of $I - \lambda\hat{H}(x)$ for all $x$ (which exists because there is only finitely many $x$). Similarly, we can define $r_{\min} > 0$ to be smaller than all the absolute values of all the eigenvalues of $I - \lambda\hat{H}(x)$ for all $x$. Then,

$$\log r_{\min} < \Lambda < \log r_{\max}. \tag{14}$$

While it is in general challenging but worthwhile to theoretically estimate the exponent (Crisanti et al., 2012; Pollicott, 2010; Jurga & Morris, 2019), it is a quantity that can be easily experimentally studied, and studying how Lyapunov exponents change with respect to hyperparameters of training algorithms.

Now, we give two quantitative estimates of $\Lambda$. This discussion also implies a sufficient but weak condition for a general type of multidimensional dynamics to converge in probability. Let $h^*(x)$ be the largest eigenvalue of $\hat{H}(x)$ and assume that $1 - h^*(x) > 0$ for all $x$. Then, the following condition implies that $\theta \to_p 0$: $\mathbb{E}_x[\log|1 - \lambda h^*(x)|] < 0$, which mimics the condition we found for rank-1 systems. An alternative estimate relies on the common diagonal approximation of the Hessian (LeCun et al., 1989). If $\hat{H}$ has rank $d$, this reduces the problem to $d$ separated rank-1 dynamics, and Theorem 1 again gives the exact solutions in each subspace. Numerical evidence shows that the diagonal approximation quite accurately predicts the onset of low-rank behavior in training (see Section 5.2 and Appendix A.5).

---

[3]If $\bar{H}$ is a $d$-by-$d$ matrix, a well-known fact is that the initialization-dependent Lyapunov exponent takes at most $d$ distinctive values.

Another potential concern is whether this theorem is trivial for SGD at a high dimension in the sense that $\Lambda$ could be identically zero independent of the dataset. One can show that the Lyapunov exponent is generally nonzero under a mild condition. Let $\mathbb{E}[\hat{H}]$ be full rank. By definition,

$$\Lambda = \lim_{t \to \infty} \frac{1}{t} \mathbb{E}\left[ \log \frac{\theta_0^T F F^T \theta_0}{\|\theta_0\|^2} \right] = -\frac{2\lambda \theta_0^T H \theta_0}{\|\theta_0\|^2} + O(\lambda^2),$$

where $F = \prod_j^t (I - \lambda \hat{H}_{i_j})$, and we have assumed that $\theta^* = 0$ for notational simplicity. Therefore, as long as $\lambda$ is sufficiently small, the sign of the Lyapunov exponent is opposite to the sign of the eigenvalues of $H$. This proves something quite general for SGD at an interpolation minimum: with a small learning rate, the model converges to the minimum exponentially fast, in agreement with common analysis in the optimization literature. See Figure 2-right for the numerical computation of Lyapunov exponents of a matrix factorization problem and the corresponding phases.

## 5 IMPLICATIONS AND EXPERIMENTS

We first present a synthetic example to compare the notations of moment stability and probabilistic stability. We then apply the theory to understand the difficulty in initializing a deep neural network with a large learning rate. Experimental details are presented in Section A.

### 5.1 COMPARISON WITH MOMENT STABILITY ON A SYNTHETIC DATASET

We start with an instructive case study that directly compares the attractivity of fixed points predicted by the probabilistic stability and the $L_p$ stability. We consider a two-layer network with a single hidden neuron with the swish activation function: $f(w, u, x) = u \times \mathrm{swish}(wx)$, where $\mathrm{swish}(x) = x \times \mathrm{sigmoid}(x)$. We generate 100 data points $(x, y)$ as $y = 0.1\mathrm{swish}(x) + 0.9\epsilon$, where both $x$ and $\epsilon$ are sampled from normal distributions. See Figure 3 for an illustration of the training loss landscape. There are two local minima: solution A at roughly $(-0.7, -0.2)$ and solution B at $(1.1, -0.3)$. Here, the solution with better generalization is A because it captures the correct correlation between $x$ and $y$ when $x$ is small. Solution A is also the sharper one; its largest Hessian eigenvalue is roughly $h_a = 7.7$. Solution B is the worse solution; it is also the flatter one, with the largest Hessian value being $h_b = 3.0$. There is also a saddle point C at $(0, 0)$, which performs significantly better than B and slightly worse than A in terms of generalization. To make theoretical predictions, We analytically compute the local Hessian matrix of the loss function (which is possible for this 2d problem). We compute its largest eigenvalue for different minibatches and apply Theorem 1 as a theoretical estimate of the stability and the critical learning rate above which the stability is lost. For the baseline $L_2$ stability, it is never stable for saddles by Proposition 2 and for local minima, its stability approximately given by Theorem 1 in Wu et al. (2018), which can be computed from the distribution of Hessian.

If we initialize the model at A, $L_2$ stability theory would predict that as we increase the learning rate, the solution moves from the sharper minimum A to the flatter minimum B when SGD loses $L_2$ stability in A; the model would then lose total stability once SGD becomes $L_2$-unstable at B. As shown by the red arrows in Figure 3. In contrast, probabilistic stability predicts that SGD will move from A to C as C becomes attractive and then lose stability, as the black arrows indicate.[4] See the right panel of the figure for the comparison with the experiment for the model's generalization performance. The dashed lines in the middle and right panels show the predictions of the $L_2$ stability and probabilistic theories, respectively. We see that the probabilistic theory predicts both the error and the place of transition right, whereas $L_2$ stability neither predicts the right transition nor the correct level of performance. If we initialize at B, the flatter minimum, $L_2$ stability theory would predict that the solution will only have one jump from B to divergence as we increase the learning rate. Thus, from $L_2$ stability, SGD would have roughly the performance of B until it diverges, and having a large learning rate will not help increase the performance. In sharp contrast, the probabilistic stability predicts that the solution will have two jumps: it stays at B for a small $\lambda$ and jumps to C as it becomes attractive at an intermediate learning rate. The model will ultimately diverge if C

---

[4] A limitation of stability-based analysis is that the theory can only tell where the algorithm will not converge. It functions thus like a worst-case guarantee. However, there is a rough rule-of-thumb for where the SGD might go when there are multiple stable solutions: when unstable, SGD makes updates based on the local noise. This makes it more likely for SGD to travel to the closest stable solution from its initial location, so C is the more likely place to converge, and this intuition indeed agrees with the experiment.

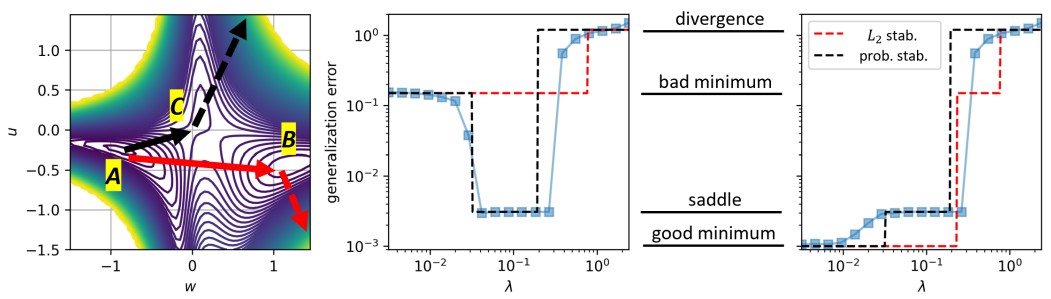

Figure 3: Dynamics of SGD in a synthetic task. **Left**: The landscape of a two-layer network with the swish activation function (Ramachandran et al., 2017). The black arrow corresponds to the experimental trajectory and the prediction of probabilistic stability, while the red arrow corresponds to the (false) prediction of the $L_2$ stability. **Middle, Right**: the generalization performance of the model for different learning rates. **Middle**: Initialized at solution B, SGD first jumps to C and then diverges. **Right**: Initialized at A, SGD also jumps to C and diverges. In both cases, the behavior of SGD agrees with the prediction of the probabilistic stability instead of the $L_2$ stability. Instead of jumping between local minima, SGD, at a large learning rate, transitions from minima to saddles.

loses stability. Thus, our theory predicts that the model will first have a bad performance, then show a better performance at an intermediate learning rate, and finally diverge. See Figure 3-mid.

### 5.2 Phases of Learning

An important observation is that almost all neural networks are initialized close to a Type-II saddle. Let $f(x) = W^{(D)}\sigma(W^{(D-1)}\sigma(...\sigma(W^{(1)}x + b^{(1)})) + b^{(D-1)})$ be a generic neural network with depth $D$ and activation $\sigma$, and $W^{(D)} \in \mathbb{R}^{d_y \times d_{D-1}}$, $W^{(i)} \in \mathbb{R}^{d_i \times d_{i-1}}$, and $W^{(1)} \in \mathbb{R}^{d_1 \times d_x}$ are of arbitrary dimensions that match the input and output dimensions. Here, $\sigma(x) = c_0 x + O(x^2)$ is any non-linearity that is locally linear at $x = 0$. Let $y = y(x)$ be the label of $x$. Assuming that the per-sample loss is differentiable, one can prove the following result, which is relevant for the standard small initialization of neural networks.

**Theorem 4.** *For any data distribution, loss function $\ell$, and any $\sigma$, the point $(W^{(D)}, ..., W^{(1)}, b^{(D-1)}, ..., b^{(1)}) = (0, ..., 0)$ is a Type-II saddle of $\ell(f(x), y(x))$ if it is not a local minimum.*

Now, we solve an analytical model of the initialization saddle that exhibits complex and dramatic phase transition-like behaviors as we change the learning rate of SGD. Consider the loss function $L = s\mathbb{E}_x[(\sum_i u^{(i)}\sigma(w^{(i)}x) - y)^2/2]$. We

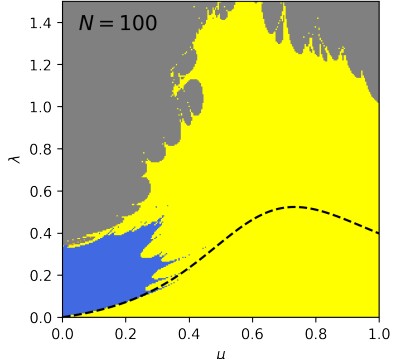

Figure 4: Phase diagram at $N = 100$. The colors indicate the phases as in Figure 2. At a finite-size, the phase boundaries have a fractal-like structure. The bottom-left of the phase diagram has a smooth boundary and is shared across almost all phase diagrams we plotted.

let $c_0 = 1$ and both $x$ and $y$ be one-dimensional. At $u, w \approx 0$, the model $u^T w$ is either rank-1 or rank-0. The rank-0 point where $u^{(i)} = w^{(i)} = 0$ for all $i$ is a saddle point as long as $\mathbb{E}[xy] \neq 0$. To leading order, the loss function takes the following form: $\ell(u, w; x, y) = -xy \sum_i^d u^{(i)}w^{(i)} + const$. For learning to happen, SGD needs to escape from the saddle point. Let us consider a simple data distribution where $xy = 1$ and $xy = a$ with equal probability. When $a > -1$, *correct* learning happens when $\text{sign}(w) = \text{sign}(u)$. We thus focus on the case of $a > -1$. The case of $a < -1$ is symmetric to this case up to a rescaling. This example is already presented in Figure 2. There are five phases of learning in this simple example:

- Ia. correct learning with prob. and norm stability ($w_t - u_t \to_{L_2} 0$, $w_t + u_t$ diverges);
- Ib. correct learning with prob. but not norm stability ($w_t - u_t \to_p 0$, $w_t - u_t \not\to_{L_2} 0$, $w_t + u_t$ diverges);
- II. incorrect learning under probabilistic stability ($w_t - u_t$ diverges, $w_t + u_t \to_p 0$);
- III. convergence to low-rank saddle point ($w_t - u_t \to_p 0$, $w_t + u_t \to_p 0$);
- IV. completely unstable ($w_t + u_t$, $w_t - u_t$ diverges in p.).

Let $B$ denote a mini-batch and $S$ be its cardinality. The following proposition precisely characterizes the phase boundaries due to SGD training.

**Proposition 2.** *For any* $w_0, u_0 \in \mathbb{R}/\{0\}$. $w_t - u_t \rightarrow_p 0$ *if and only if* $\mathbb{E}_B[\log|1 - \lambda \sum_{(x,y) \in B} xy/S|] < 0$. $w_t + u_t$ *converges to* $0$ *in probability if and only if* $\mathbb{E}_B[\log|1 + \lambda \sum_{(x,y) \in B} xy/S|] < 0$.

This shows that the phase diagram of SGD strongly depends on the data distribution, and it is interesting to explore and compare a few different settings. Now, we consider a size-$N$ Gaussian dataset. Let $x_i \sim \mathcal{N}(0, 1)$ and noise $\epsilon_i \sim \mathcal{N}(0, 4)$, and generate a noisy label $y_i = \mu x_i + (1 - \mu)\epsilon_i$. See the phase diagram for this dataset in Figure 4. We see that the phase diagram has a rich structure at a finite size. There are three rather surprising observations about the phase diagrams: (1) as $N \rightarrow \infty$, the phase diagram becomes smoother and smoother and each phase takes a connected region (cf. experiments in Appendix A.2); (2) phase II seems to disappear as $N$ becomes large; (3) the lower part of the phase diagram seems universal, taking the same shape for all samplings of the datasets and across different sizes of the dataset. This suggests that the convergence to low-rank structures can be a universal aspect of SGD dynamics, which corroborates the widely observed phenomenon of collapse in deep learning (Papyan et al., 2020; Wang & Ziyin, 2022; Tian, 2022). The theory also shows that if we fix the learning rate and noise level, increasing the batch size makes it more and more difficult to converge to the low-rank solution (see Figure 11, for example). This is expected because the larger the batch size, the smaller the effective noise in the gradient.

This discussion explains a commonly observed yet puzzling phenomenon in the initialization of neural networks – when $\lambda$ is small, increasing $\lambda$ leads to faster initialization; whereas for a large $\lambda$, increasing $\lambda$ often slows down initialization. See Figure 5 and Appendix A.7 for examples of this effect in training a deep neural network on MNIST. Also, compare this experiment with the Figure 2-Right and observe the qualitative similarity.

### 5.3 INITIALIZATION OF DEEP NEURAL NETWORKS

This subsection verifies the above discussion that neural networks often have significant problems in initiating training if the SNR in the gradient is small. We start with a controlled experiment where, at every training step, we sample input $x \sim \mathcal{N}(0, I_{200})$ and noise $\epsilon \sim \mathcal{N}(0, 4I_{200})$, and generate a noisy label $y = \mu x + (1 - \mu)\epsilon$. Note that $1 - \mu$ controls the level of the noise. Training proceeds with SGD on the MSE loss. We train a two-layer model with the architecture: $200 \rightarrow 200 \rightarrow 200$. See Figure 6 for the result, in comparison to the theoretical phase boundary between escaping and entrapment, which is estimated under the diagonal approximation. Under SGD, the model escapes from the saddle with a finite variance to the right of the dashed line and has an infinite variance to its left. In the region $\lambda \in (0, 0.2)$, this loss of the $L_2$ stability condition coincides with the condition for the convergence to the saddle. The experiment shows that the

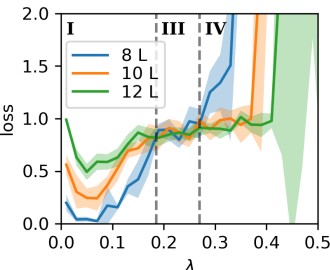

Figure 5: Normalized training loss ($\ell/\ell_{\text{init}}$) of deep fully connected $tanh$ networks after 2000 steps, as a function of learning rate $\lambda$. The curves represent the averages of five independent runs. The three phases of learning in the 8-layer case are marked by the text and the vertical dashed lines.

theoretical boundary agrees well with the numerical results. The Adam optimizer (Kingma & Ba, 2014) also have a similar phase diagram (Appendix A.6). This suggests that the effects we studied are rather universal, not just a special feature of SGD.

Then, we train independently initialized ResNets on CIFAR-10 with SGD. The training proceeds with SGD without momentum at a fixed learning rate and batch size $S = 32$ (unless specified otherwise) for $10^5$ iterations. We perform experiment with two types of noises: (a) dynamically injected label noise during every training step, where, at every step, a correct label is flipped to a random label with probability $p =$ noise and (b) a static noise where the labels are flipped at $p =$ noise before the training starts. See Figure 10 for the phase diagram of static label noise. The empirical phase diagrams are similar regardless of whether the noise is dynamical or static (where the mislabelling is fixed). Interestingly, the best generalization performance is achieved close to the phase boundary when the noise is strong. This is consistent with the observation that SGD noise has a strong regularization effect on the trained model. We also plot the sparsity of the ResNets in different layers in Figure 7, and we observe that the phase diagrams are all qualitatively similar.

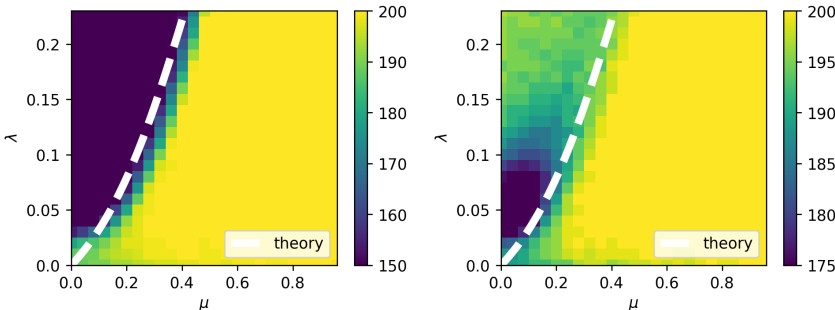

Figure 6: Convergence to low-rank solutions in nonlinear neural networks. At every training step, we sample input $x \sim \mathcal{N}(0, I_{200})$ and noise $\epsilon \sim \mathcal{N}(0, \sqrt{2}I_{200})$, and generate a noisy label $y = \mu x + (1 - \mu)\epsilon$, where $1 - \mu$ controls the level of the noise. We compute the rank of the second layer of the weight matrix after training. **Left**: Linear network. **Right**: tanh network. The white dashed line shows the theoretical prediction of the appearance of low-rank structure computed by numerically integrating the condition in Proposition 2.

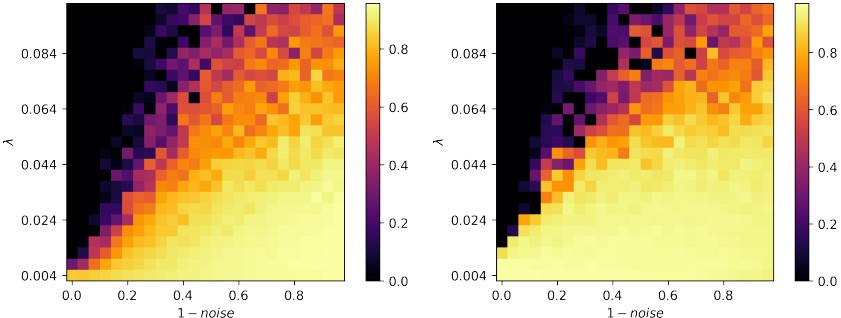

Figure 7: Density $(1 - sparsity)$ of the convolutional layers in a ResNet18, when there is static noise (mislabeling) in the training data. Here, we show the number of sparse parameters in the two of the largest convolutional layers, each containing roughly one million parameters in total. The figures respectively show layer1.1.conv2 (upper left), layer2.1.conv2 (upper right).

Also, see Appendix A.4 for the experiment with a varying batch size. These experiments show that Type-II saddles are indeed a major obstacle in the initial phase of training. This result may also explain why the warmup is needed for training many neural networks. Training transformers is only successful if one starts with a small learning rate and increases it slowly toward a maximal value. If the training starts at a large learning rate, one often observes that the model becomes trapped in a low-capacity state soon after initialization Vaswani et al. (2017).

## 6 DISCUSSION

We have shown that special attention should be paid to the study of Type-II saddle points in neural networks. To this purpose, we demonstrate that the probabilistic stability serves as an essential notion for understanding the attractivity of Type-II saddle points as no moment-based notions of stability can be used to study when a saddle point becomes attractive. These effects are only present for SGD and not for GD, implying that the algorithmic regularization due to SGD is qualitatively different from that of GD. Our result also sheds light on how a solution in the loss landscape is chosen by SGD. In the probabilistic-stability perspective, SGD performs a selection between converging to saddles and to local minima, not between sharp minima and flat ones. A main limitation of our work is that we only focused on the local dynamics close to stationary points, and it would be important to link it to the global learning dynamics. Depending on how one views the problem of Type-II saddles, there are two interesting future directions. If we regard the Type-II saddles as major obstacles to optimization, an important future problem is to design algorithms to better escape from them or to remove them from the loss landscape. Alternatively, the Type-II saddles can function as effective capacity constraints on the models, and one might also be interested in leveraging them to better regularize neural networks.

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

## A  EXPERIMENTAL CONCERNS

### A.1  EXPERIMENTAL DETAIL FOR FIGURE 2-RIGHT

The experiment is performed for a two-dimensional system whose dynamics is specified in Eq. (7). The expectation of the Hessian $\mathbb{E}[\hat{H}]$ is chosen to be $\text{diag}(0.1, -0.1)$, while the noise is generated via a normal $2 \times 2$ random matrix $M_{\text{noise}}$. The noisy Hessian is obtained as

$$\hat{H} = \mathbb{E}[\hat{H}] + M_{\text{noise}} + M_{\text{noise}}^T, \tag{15}$$

and one can verify that such $\hat{H}$ is symmetric and consistent with our choice of $\mathbb{E}[\hat{H}]$. The initial state is sampled from a unit circle. The dynamics stops at time step $t$, and the Lyapunov exponent is calculated as $\frac{1}{t} \log ||\theta_t||$, if one of the three following conditions is satisfied: $||\theta||$ reaches the upper cutoff of $10^{100}$; $||\theta||$ reaches the lower cutoff of $10^{-140}$; the preset maximal number of steps of $5000$ is reached. For each learning rate, the Lyapunov is obtained as the average of the results collected in $800$ independent runs.

### A.2  PHASES OF FINITE-SIZE DATASETS

See Figure 8.

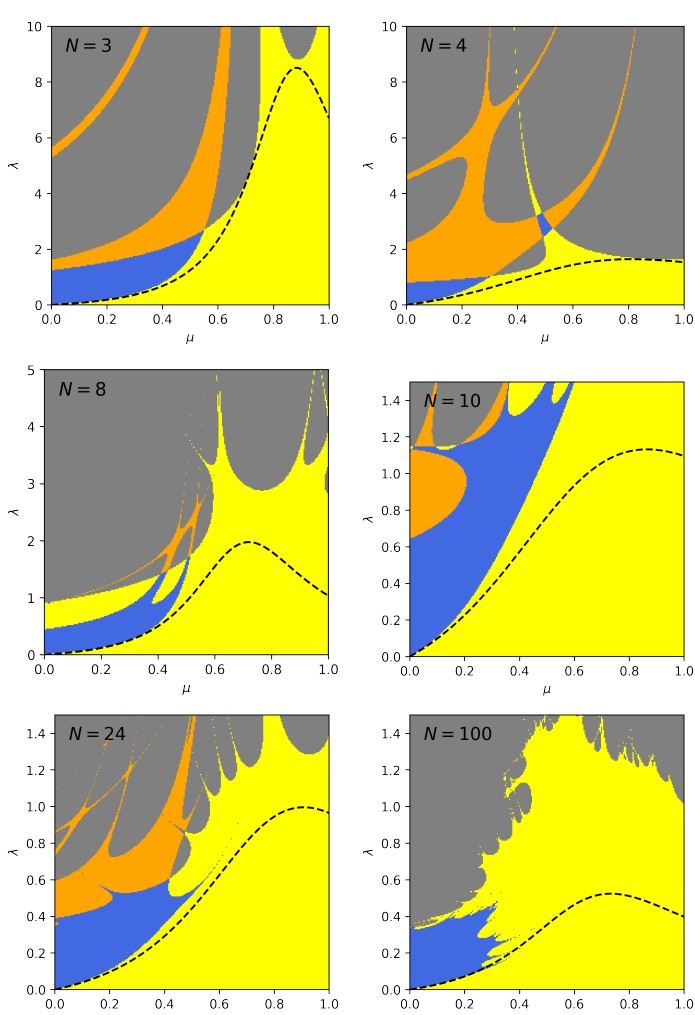

Figure 8: **Phase diagrams of SGD stability for finite-size dataset**. The data sampling is the same as in Figure 9. From upper left to lower right: $N = 3, 4, 8, 10, 24, 100$. As the dataset size tends to infinity, the phase diagram converges to that in Figure 9. The lower left parts (strong noise and weak data correlation) of all the phase diagrams look similar, suggesting a universal structure.

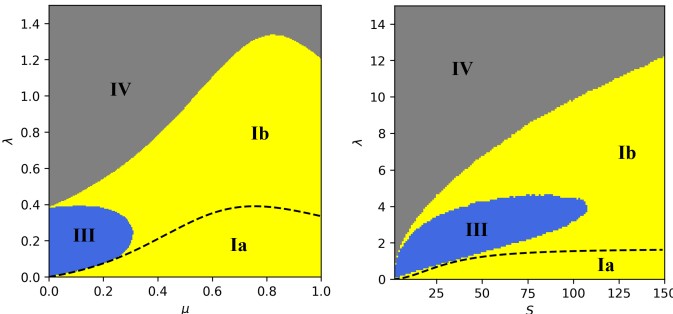

Figure 9: **Phase diagrams of SGD stability**. The definitions of the phases are the same as those in Figure 2. We sample a dataset of size $N$ such that $x \sim \mathcal{N}(0,1)$ and noise $\epsilon \sim \mathcal{N}(0,4)$, and generate a noisy label $y = \mu x + (1 - \mu)\epsilon$. Left: the $\lambda - \mu$ (noise level) phase diagram for $S = 1$ and $N = \infty$. Right: The $\lambda - S$ (batch size) phase diagram for $\mu = 0.06$ and $N = \infty$.

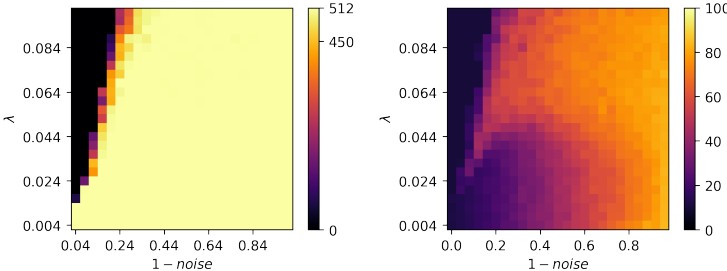

Figure 10: Rank (**left**) and test accuracy (**right**) of the ResNet18 trained in a data set with static noise. The transition of rank has a clear boundary. The model has a full rank but random-guess level performance for large noise and small learning rates. Here, *noise* refers to the probability that the data point is mislabeled.

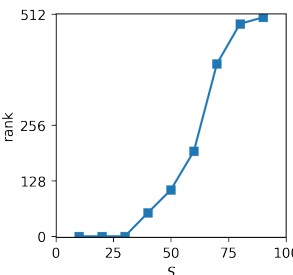

Figure 11: The rank of the penultimate-layer representation of Resnet18 trained with different levels of batch sizes. In agreement with the phase diagram, the model escapes from the low-rank saddle as one increases the batch size.

### A.3    RESNET18 WITH STATIC LABEL NOISE

See Figure 10, where the labels of the training set is corrupted at a fixed probability to a different fixed label during training. We see that the problem of initialization remains and features a similar phase boundary.

### A.4    EFFECT OF CHANGING BATCH SIZE ON RESNET18

See Figure 11.

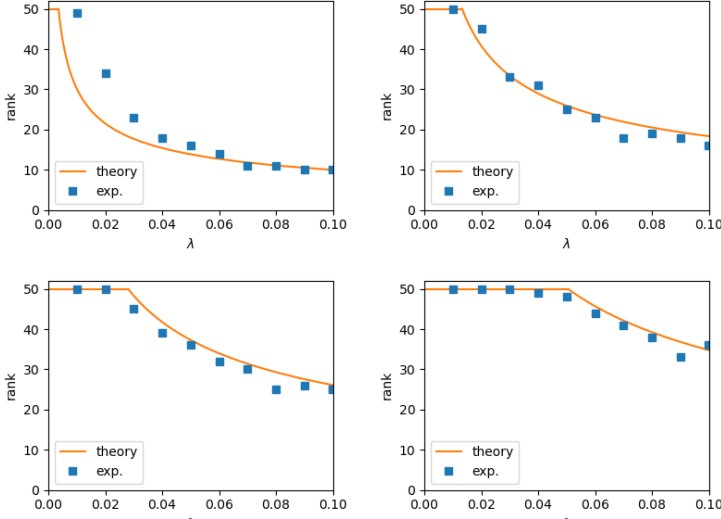

Figure 12: Rank vs learning rate in a vanilla matrix factorization problem for, from upper left to lower right, $\mu = 0.05,\ 0.15,\ 0.25,\ 0.35$. The theoretical curve is from the diagonal approximation where each subspace of the model collapses at the critical learning rate $\lambda = -2\frac{\mathbb{E}[h(x)]}{\mathbb{E}[h^2(x)]}$.

### A.5 Diagonal Approximation

Here, we compare the empirical rank of the solution with the diagonally approximated critical learning rates obtained in Eq. (11). See Figure 12. The experiment is run on a two-layer fully connected linear network: $50 \to 50 \to 50$, which is equivalent to a matrix factorization problem. The model is initialized with the standard Kaiming init. The dataset we consider is one with a sparse but full-rank signal.

Let $\odot$ denote the Hadamard product. The input data is generated as $x = m \odot X$, where $X \sim \mathcal{N}(0, I_{50})$ and $m$ is a random mask where a random element is set to be 1, and the rest is zero. The labels $Y$ are generated as $Y = \mu x + (1 - \mu)(m \odot \epsilon)$, where $\epsilon \sim \mathcal{N}(0, 2\mathrm{diag}(0.01, 0.05, ...., 2.01))$ is the noise.

### A.6 Experiment with Adam

We note that the phenomena we studied is not just a special feature of the SGD, but, empirically, seems to be a universal feature of first-order optimization methods that rely on minibatch sampling. Here, we repeat the experiment in Figure 9. We train with the same data and training procedure, except that we replace SGD with Adam (Kingma & Ba, 2014), the most popular first-order optimization method in deep learning. Figure 13 shows that similarly to SGD, Adam also converges to the low-rank saddles in similar regions of learning rate and $\mu$.

### A.7 Trapped at initialization

Here, we train fully connected $tanh$ network for MNIST image classification and demonstrate that SGD can be trapped at the initialization for a large enough learning rate. The image data are flattened into 784-dimensional vectors, which are then passed into a $784 \to 128 \to ... \to 128 \to 10$ fully connected neural network with $tanh$ activation functions. The experiment is performed with different numbers of hidden layers, all of the same size, $128 \times 128$. The parameters are initialized uniformly in the interval $[-1/\sqrt{\texttt{in\_features}}, 1/\sqrt{\texttt{in\_features}}]$. The network is trained using SGD of a batch size one and a constant learning rate $\lambda$, and training stops after the first 2000 steps. To improve stability with large $\lambda$, we implement gradient clipping. The evolution of loss is shown in Figure 5. The plotted loss is estimated using the median loss of the last 20 steps, normalized by

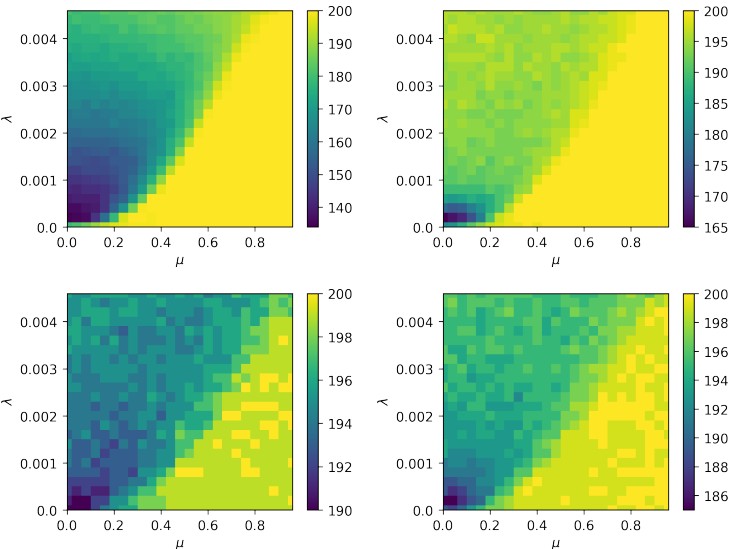

Figure 13: Rank of the converged solution for two-layer linear (upper left), tanh (upper right), relu (lower left) and swish (lower right) models.

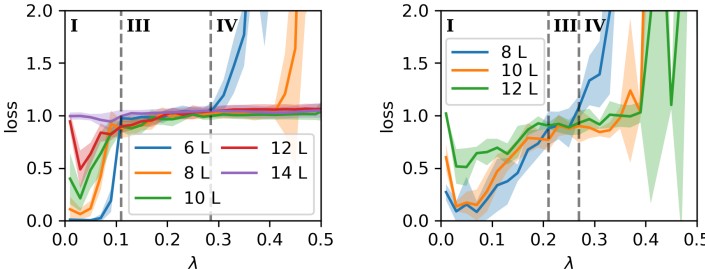

Figure 14: Normalized training loss as a function of learning rate of deep fully connected networks with (left) linear activation functions and (right) ReLU. The curves represent averages of five independent runs. The three phases of learning in the 6-layer (8-layer resp.) case with linear activation functions (ReLU resp.) are marked by the text and the vertical dashed lines.

the initial loss, and averaged over five independent runs. The shaded area indicates the $1$-$\sigma$ error bar of the normalized loss.

If the normalized loss is 1, one can conclude that the performance in the training set has not changed. Thus, the final state is close to the initial state in the projected space. For the 8-layer case, the training-loss ratio decays quickly towards zero for small learning rates but rises rapidly at $\lambda \approx 0.18$, then stops at 1 before diverging at larger learning rates. The phases of learning in the 8-layer case is marked in the figure. For the 10-layer case, the normalized loss follows a similar trend, but the phase transition is less pronounced. As shown in Figure 14, the same experiment is also performed for fully connected deep linear networks and fully connected ReLU networks. The results agree qualitatively with those for $tanh$ networks.

Also, the Kaiming uniform initialization (He et al., 2015) remains within the neighborhood of the saddle point at the origin. To demonstrate this, we plot the loss along interpolated weights, spanning from the origin to 1.5 times the initialization weights. This curve reflects the loss as measured along the initialization direction starting from the origin. We consider two problems: the CIFAR-10 image classification using ResNet18 and uncovering the rule generating synthetic data. For the second task, the data pairs $(x_i, y_i)$ are generated by the relation $y_i = v \cdot \tanh(x_i) + \epsilon_i$, where $x_i \in \mathbb{R}^{10}$ are independent standard normal vectors, $\epsilon_i \in \mathbb{R}$ is a Gaussian noise with standard

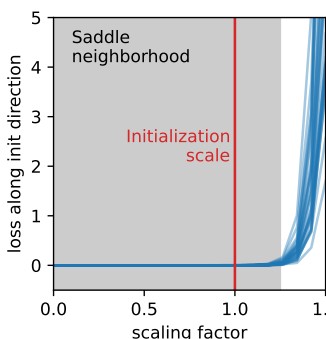 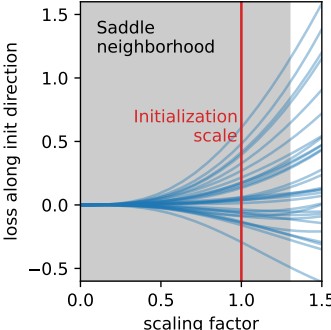

Figure 15: Interpolating and extrapolating the loss function along the direction of initialization. Each curve represents the loss computed with an independent initialization, rescaled from 0 to 1.5. The shaded region indicates the vicinity of the saddle point at the origin. (Left) ResNet18 on CIFAR-10. (Right) MLP with $\tanh$ activation on synthetic data.

deviation 0.03, and $v \in \mathbb{R}^{10}$ is a normal vector shared across all data points. The MLP contains one hidden layer with ten neurons. Figure 15 shows the loss curves for 30 independent initializations. The $x$-axis represents the factor multiplied by the weights, with the initialization locates at 1. For ResNet18, the loss remains close to zero until factor reaches 1.2. This aligns the conclusion we get during the proof of Theorem 4, which asserts that the Hessian at the original is trivial. It also demonstrates that the initialization consistently lies within the flat region around the origin, where the linearized dynamics provides a good approximation. For MLP, the loss is less flat, and the origin is clearly a saddle point. Most of the thirty curves are monotonic, with two or three exhibiting a local minimum. We truncate the neighborhood of the saddle at the closest local minimum, confirming that the initialization remains within the origin's vicinity. These findings support our claim that the initialization is near a Type-II saddle.

## B  PROOFS

### B.1  SETTING FOR FIGURE 1

The following derivation gives the construction of the Type-I and Type-II saddle points in Figure 1. Let us first study the case when there is a single neuron. The network is

$$f(x, \theta) = \sum_i u_i \sigma(w_i^T x), \tag{16}$$

where $\sigma = \text{ReLU}$ is the activation, and $x$ contains an element of 1 to include the effect of having a bias.

The loss function is $L = \frac{1}{N} \sum_x \ell(x)$, where

$$\ell(x) = (f(x) - y(x))^2 \tag{17}$$

Without loss of generality, we can write $\mathbb{E}_x$ in the place of $\frac{1}{N} \sum_x$, where the expectation is taken over the training set. Let $\sigma = \text{ReLU}$. We have that $\sigma'' = 0$. The gradients are

$$\nabla_u \ell(x) = \mathbb{1}_> w^T x (u w^T x - y), \tag{18}$$

$$\nabla_w \ell(x) = \mathbb{1}_> u (u w^T x - y) x. \tag{19}$$

where we have used $\mathbb{1}_>$ as an indicator function of the event $w^T x \geq 0$. The Hessian is

$$H(x) = \begin{bmatrix} \mathbb{1}_> (w^T x)^2 & \mathbb{1}_> w^T x u x + \mathbb{1}_> (u(w^T x) - y) x^T \\ \mathbb{1}_> w^T x u x + \mathbb{1}_> (u(w^T x) - y) x & \mathbb{1}_> u^2 x x^T \end{bmatrix} \tag{20}$$

There are two cases: (1) $u = 0$ and (2) $u \neq 0$.

For the case $u = 0$, we have that $\nabla_w L = 0$. For $\nabla_u L = 0$, we must have

$$\nabla_u L = \mathbb{E}[\mathbb{1}_> w^T x (-y)] = 0 \tag{21}$$

which is solved by any $w$ such that $w^T \mathbb{E}[\mathbb{1}_> x y] = 0$. These solutions are all saddle points. To see this, note that the sample Hessian is:

$$H(x) = \begin{bmatrix} \mathbb{1}_> (w^T x)^2 & -\mathbb{1}_> y x^T \\ -\mathbb{1}_> y x & 0 \end{bmatrix}, \tag{22}$$

The expected Hessian is:

$$\mathbb{E}[H(x)] = \begin{bmatrix} \mathbb{E}[\mathbb{1}_> (w^T x)^2] & -\mathbb{E}[\mathbb{1}_> y x^T] \\ -\mathbb{E}[\mathbb{1}_> y x] & 0 \end{bmatrix}, \tag{23}$$

which is a rank-2 matrix whose eigenproblem is solved by eigenvalues

$$\lambda_\pm = \frac{1}{2} \left( \mathbb{E}[\mathbb{1}_> (w^T x)^2] \pm \sqrt{\mathbb{E}[\mathbb{1}_> (w^T x)^2]^2 + \|\mathbb{E}[\mathbb{1}_> y x^T]\|^2} \right), \tag{24}$$

and eigenvectors:

$$v_\pm = (\lambda_\pm, -\mathbb{E}[\mathbb{1}_> y x^T]). \tag{25}$$

Obviously, $\lambda_- < 0$, and these solutions are strict saddle points, except for the directions of $w$ that $\mathbb{E}[\mathbb{1}_> y x^T] = 0$.

Now, a crucial observation is that the saddle points given by this condition can be divided into two classes with dramatically different properties under a stochastic optimization algorithm: (I) $w \neq 0$ and (II) $w = 0$. For the type I saddle, we have that the sample-wise gradient is

$$\nabla_u \ell = \mathbb{1}_> w^T x (u w^T x - y), \tag{26}$$

which is a random variable that does not vanish in general. This random noise facilitates the escape from the saddle.

For the type II, we have that with probability 1,

$$\nabla_u \ell = 0, \tag{27}$$

which signals a vanishing noise, the lack of which means that this type of saddle is difficult to escape. Alternatively, let $u = u_0 + \Delta u$ and $w = w_0 + \Delta w$ denote a small deviation from the saddle $(u_0, w_0)$. Then,

$$\nabla_u \ell(x) = \begin{cases} \Theta(1) & \text{if } w_0 \neq 0; \\ O(\|\Delta w\|) & \text{if } w_0 = 0. \end{cases} \tag{28}$$

$$\nabla_w \ell(x) = O(\|\Delta u\|). \tag{29}$$

## B.2 PROOF OF THEOREM 1

*Proof.* Consider Eq. (7):

$$\theta_{t+1} = \theta_t - \lambda \hat{H}(x)(\theta_t - \theta^*). \tag{30}$$

Defining $w_t = \theta_t - \theta^*$, this equation can be written as

$$w_{t+1} = w_t - \lambda \hat{H}(x) w_t, \tag{31}$$

which is mathematically equivalent to the case when $\theta^* = 0$. Therefore, without loss of generality, we write the dynamics in the form of Eq. (31) in this proof and the rest of the proofs.

Now, when $\hat{H} \propto nn^T$ is rank-1, we can multiply $n^T$ from the left on both sides of the dynamics to obtain

$$n^T w_{t+1} = n^T w_t - \lambda h(x) n^T w_t. \tag{32}$$

The dynamics thus becomes one-dimensional in the direction of $n^T$.

Let $h_t$ denote the eigenvalue of the Hessian of the randomly sampled batch at time step $t$. The dynamics in Eq. (9) implies the following dynamics

$$\|n^T w_{t+1}\| / \|n^T w_t\| = |1 - \lambda h_t|, \tag{33}$$

which implies

$$\|n^T w_{t+1}\| / \|n^T w_0\| = \prod_{\tau=1}^{t} |1 - \lambda h_\tau|. \tag{34}$$

We can define auxiliary variables $z_t := \log(\|n^T w_{t+1}\| / \|n^T w_0\|) - m$ and $m := \mathbb{E}[\log(\|n^T w_{t+1}\| / \|n^T w_0\|)] = t\mathbb{E}_x[\log|1 - \lambda h_t|]$. Let $\epsilon > 0$. We have that

$$\mathbb{P}(\|n^T w_t\| < \epsilon) = \mathbb{P}(\|n^T w_0\| e^{z_t + m} < \epsilon) \tag{35}$$

$$= \mathbb{P}\left(\frac{1}{t} z_t < \frac{1}{t}(\log \epsilon / \|n^T w_0\| - m)\right) \tag{36}$$

$$= \mathbb{P}\left(\frac{z_t}{t} < -\mathbb{E}_x \log|1 - \lambda h_t| + o(1)\right). \tag{37}$$

By the law of large numbers, the left-hand side of the inequality converges to 0, whereas the right-hand side converges to a constant. Thus, we have, for all $\epsilon > 0$,

$$\lim_{t \to \infty} \mathbb{P}(\|n^T w_t\| < \epsilon) = \begin{cases} 1 & \text{if } m < 0; \\ 0 & \text{if } m > 1. \end{cases} \tag{38}$$

This completes the proof. $\qquad \square$

## B.3 PROOF OF PROPOSITION 2

*Proof.* Part 2 of the proposition follows immediately from Proposition 3, which we prove below. Here, we prove part 1.

It suffices to consider a dataset with two data points for which $h(x_1) = 1/\lambda$ and $h(x_2) = c_0$, where each data point is sampled with equal probability. Let $c_0$ be such that

$$|1 - \lambda c_0|^p > \frac{1}{2}. \tag{39}$$

Now, we claim that this dynamics converges to zero in probability. To see this, note that

$$\|z_{t+1}\| = \begin{cases} \|z_t\| |1 - \lambda/\lambda| = 0 & \text{with probability } 0.5; \\ \|z_t\| |1 - \lambda c_0| & \text{with probability } 0.5. \end{cases} \tag{40}$$

Therefore, at time step $t$, $\mathbb{P}(z_t = 0) \geq 1 - 2^{-t}$, which converges to 0. This means that $z_t$ converges in probability to 0.

Meanwhile, the $p$-norm is

$$\mathbb{E}[\|z_{t+1}\|^p] = \frac{1}{2}\mathbb{E}[\|z_t\|^p]|1 - \lambda c_0|^p \tag{41}$$

$$\propto \frac{1}{2^t}|1 - \lambda c_0|^{pt} \to 0. \tag{42}$$

The convergence to zero follows from the construction that $|1 - \lambda c_0|^p > \frac{1}{2}$. This completes the proof. $\qquad\square$

**Proposition 3.** *(No convergence in $L_p$.) For every strict saddle point $\theta^*$, there exists an initialization $\theta_0$ such that for any $\lambda \in \mathbb{R}_+$ and distance function $f(\cdot, \theta^*)$, $\theta^*$ is unstable in $f$.*

*Proof.* This problem is easy to prove when $\theta$ is one-dimensional. For a high-dimensional $\theta$, the dynamics of SGD is

$$\theta_{t+1} = (I - \lambda \hat{H}_t)\theta_t. \tag{43}$$

Note that the expected value of $\theta_t$ is the same as the gradient descent iterations:

$$\mathbb{E}[\theta_{t+1}] = (I - \lambda \mathbb{E}[\hat{H}])\mathbb{E}[\theta_t] = (I - \lambda \mathbb{E}[\hat{H}])^t \theta_0, \tag{44}$$

which diverges if $\theta_0$ is in one of the escape directions of $\mathbb{E}[\hat{H}]$, which exist by the definition of strict saddle points.

Taking the $f-$distance of both sides and taking expectation, we obtain

$$\mathbb{E}[f(\theta_t, \theta^*)] \geq f(\mathbb{E}[\theta_t], \theta^*) \tag{45}$$

$$= f\left((I - \lambda \mathbb{E}[\hat{H}])^t \theta_0, \theta^*\right) \not\to 0. \tag{46}$$

The first line follows from the fact that the distance function is convex by definition, and so one can apply Jensen's inequality.

Therefore, as long as $\theta_0$ overlaps with the concave directions of $\mathbb{E}[\hat{H}]$, the argument of $f$ diverges, which implies that the distance function converges to a nonzero value. The expected value of $\theta_t$ is just the gradient descent trajectory, which diverges for any strict saddle point.

By definition, $\mathbb{E}[\hat{H}]$ contains at least one negative eigenvalue, and so the directions that do not overlap with this direction are strict linear subspaces with dimensions lower than the the total available dimensions. This is a space with Lesbegue measure zero. The proof is complete. $\qquad\square$

### B.4 PROOF OF THEOREM 3

Let us first state the Furstenberg-Kesten theorem.

**Theorem 5.** *(Furstenberg-Kesten theorem) Let $X_1$, $X_2$, $X_3$, ... be independent random square matrices drawn from a metrically transitive time-independent stochastic process and $\mathbb{E}[\log_+ \|X^1\| < \infty]$, then*[5]

$$\lim_{n\to\infty} \frac{1}{n} \log \|X_1 X_2 ... X_n\| = \lim_{n\to\infty} \mathbb{E}\left[\frac{1}{n} \log \|X_1 X_2 ... X_n\|\right] \tag{47}$$

*with probability 1, where $\|\cdot\|$ denotes any matrix norm.*

Namely, the Lyapunov exponent of every trajectory converges to the expected value almost surely. Essentially, this is a law of large numbers for the Lyapunov exponent.

Now, we present the proof of Theorem 3.

*Proof.* First of all, we define $m_t = \log \|\theta_t - \theta^*\|$ and $z_t = m_t - \mathbb{E}[m_t]$. By definition, we have

$$\mathbb{P}(g_t < \epsilon) = \mathbb{P}(e^{z_t + m_t} < \epsilon) \tag{48}$$

$$= \mathbb{P}\left(\frac{1}{t}(z_t + \mathbb{E}[m_t]) < \frac{1}{t}\log\epsilon\right) \tag{49}$$

$$= \mathbb{P}\left(\frac{1}{t}(z_t + \mathbb{E}[m_t]) < o(1)\right). \tag{50}$$

---

[5]$\log_+(x) = \max(\log x, 0)$.

We can lower bound this probability by

$$\mathbb{P}\left(\frac{1}{t}(z_t + \mathbb{E}[m_t]) < o(1)\right) \geq \mathbb{P}\left(\frac{1}{t}\max_{\theta_0}(z_t + \mathbb{E}[m_t]) < o(1)\right). \tag{51}$$

By the definition of SGD, we have

$$\frac{1}{t}\max_{\theta_0}(z_t(\theta_0) + \mathbb{E}[m_t]) = \frac{1}{t}\max_{\theta_0}\log\left\|\prod_i^t (I - \lambda\hat{H}_i)(\theta_t - \theta_0)\right\|. \tag{52}$$

By the Furstenberg-Kesten theorem (Furstenberg & Kesten, 1960), this quantity converges to the constant $\Lambda = \lim_{t\to\infty}\mathbb{E}[m_t]/t \in \mathbb{R}$ almost surely. Namely, $z_t/t$ converges to 0 for almost every SGD trajectory.

Thus, for every $\epsilon$, if $\Lambda < 0$, Eq. (50) can be bounded as

$$\lim_{t\to\infty}\mathbb{P}(g_t < \epsilon) = \mathbb{P}(\Lambda < 0) = 1. \tag{53}$$

Because $\Lambda$ is a constant, we have that if $\Lambda < 0$, all trajectories from all initialization converge to 0. This finishes the first part of the proof. For the second part, simply let $z_t$ be the trajectory starting from the trajectory that achieves the maximum Lyapunov exponent. Again, this dynamics escapes with probability 1 by the Furstenberg-Kesten theorem. The proof is complete. $\square$

## B.5 PROOF OF THEOREM 4

*Proof.* With the linear approximation of activation function $\sigma(x) \approx c_0 x$, the neural network takes the forms of $f(x) = c_0^D \prod_{i=1}^{D} W^{(i)} x + \sum_{i=1}^{D-1} c_0^{(D-i)} \prod_{j=i+1}^{D} W^{(j)} b^{(i)}$. The gradient of the loss $\ell(f(x), y(x))$ is thus

$$\nabla_\theta \ell(f(x), y(x)) = (\nabla_{f(x)}\ell)^T \nabla_\theta f(x). \tag{54}$$

For layer $i$, the gradient

$$\nabla_{W^{(i)}} f(x) \propto c_0^D \left(\prod_{j=1}^{i-1} W^{(j)} x\right)\left(\prod_{j=i+1}^{D} W^{(j)}\right) + \sum_{k=1}^{i-1} c_0^{(D-k)}\left(\prod_{j=k+1}^{i-1} W^{(j)} b^{(k)}\right)\left(\prod_{j=i+1}^{D} W^{(j)}\right); \tag{55}$$

$$\nabla_{b^{(i)}} f(x) \propto c_0^{(D-i)} \prod_{j=i+1}^{D} W^{(j)}. \tag{56}$$

As there is no constant term in the gradient, the gradient vanishes at $\theta = 0$ for all $x$, and so is the projection of the gradient. Thus, $\theta = 0$ is a Type-II saddle, under the assumption that $\theta = 0$ is not a local minimum. $\square$

## B.6 PROOF OF PROPOSITION 2

*Proof.* We consider the dynamics of SGD around a saddle:

$$\ell = -\chi \sum_i u_i w_i, \tag{57}$$

where we have combined $\frac{1}{S}\sum_{(x,y)\in B} xy$ into a single variable $\chi$. The dynamics of SGD is

$$\begin{cases} w_{i,t+1} = w_{i,t} + \lambda\chi u_{i,t}; \\ u_{i,t+1} = u_{i,t} + \lambda\chi w_{i,t}. \end{cases} \tag{58}$$

Namely, we obtain a set of coupled stochastic difference equations. Since the dynamics is the same for all values of the index $i$, we omit $i$ from now on. This dynamics can be decoupled if we consider two transformed parameters: $h_t = w_t + u_t$ and $m_t = w_t - u_t$. The dynamics for these two variables is given by

$$\begin{cases} h_{t+1} = h_t + \lambda\chi h_t; \\ m_{t+1} = m_t - \lambda\chi m_t. \end{cases} \tag{59}$$

We have thus obtained two decoupled linear dynamics that take the same form as that in Theorem 1. Therefore, as immediate corollaries, we know that $h$ converges to 0 if and only if $\mathbb{E}_B[\log|1+\lambda\chi|] < 0$, and $m$ converges to 0 if and only if $\mathbb{E}_B[\log|1-\lambda\chi|] < 0$.

When both $h$ and $m$ converge to zero in probability, we have that both $w$ and $u$ converge to zero in probability. For the data distribution under consideration in section 5.2 and for batch size one, we have

$$\mathbb{E}[\log|1+\lambda\chi|] = \frac{1}{2}\log|(1+\lambda)(1+\lambda a)| \tag{60}$$

and

$$\mathbb{E}[\log|1-\lambda\chi|] = \frac{1}{2}\log|(1-\lambda)(1-\lambda a)|. \tag{61}$$

There are four cases: (1) both conditions are satisfied; (2) one of the two is satisfied; (3) neither is satisfied. These correspond to four different phases of SGD around this saddle. $\qquad\square$

