# OpenReview forum: "Type-II Saddles and Probabilistic Stability of Stochastic Gradient Descent"
_ICLR.cc/2025/Conference — Submitted to ICLR 2025_

### Official Review · Reviewer_6FcH · 2024-10-30

**Soundness:** 2
**Presentation:** 1
**Contribution:** 3
**Rating:** 3
**Confidence:** 2

**Summary:**

The authors introduce a new classification of saddle points, Type 1 and Type 2, based on the variance exhibited by the SGD process near each type of saddle point. Using this distinction, they demonstrate that Type 2 saddle points can slow down or even prevent the convergence of SGD. This leads to a better understanding of SGD’s convergence behavior in neural network optimization.

**Strengths:**

1. The distinction in type 1 and type 2 saddles is new.

2. The authors not only provide theoretical evidence that type two saddles cause problems for SGD but also provide numerical examples to underline the results.

**Weaknesses:**

While the general idea behind the classification in these two types of saddle points is intuitive and interesting the following points weaken the contribution of the paper:

1. The notation used is often not formally introduced, details are missing. Even the under lying problem and SGD scheme is not introduced.

2. Given 1, it is hard to verify the correctness of the derived equations and theorems. See questions below.

**Questions:**

**Q1:** Sec 3. line 96:  loss $\mathcal l$ not introduced. I have a hard time to understand anything if I do not know the underlying problem and the probability space and SGD scheme is also not defined.
Could you provide:
- A formal definition of the loss function $\mathcal{l}$
- A clear description of the probability space and problem setting
- A precise definition of the SGD update rule used

**Q2:** Eq. (1): Could you clarify the dimension of $\theta $? From the context, it seems to be a vector in $ \mathbb{R}^d $, but it’s unclear how a $ d $-dimensional vector could be $ \mathcal{O} $ of a real value, and whether this holds for any $ t $. I’m not familiar with this notation and am uncertain about its meaning here. Does this equation apply to any saddle point, and how is it linked to an expected gradient norm of zero? Additionally, when you refer to “$\lVert \Delta \theta \rVert$ as the Euclidean distance from the saddle,” do you mean $\lVert \theta_t - \theta \rVert $, or is it some quantity independent of $t$?


**Q3:** Following Q1 and Q2, I’m having some difficulty understanding the derivation of the different saddle types. Could you provide a more rigorous derivation of the saddle types. Are there other types not covered by this classification? Could you explain why only these two types are considered, if that is indeed the case.

**Q4:** Could you clarify what $P$ denotes the projection of— is it $\theta$ or $H$? Based on Definition 1, it seems likely that $\theta$ is intended, but this is just my assumption.

**Q5:** How does Def. 1 imply eq. (3)? And does property (3) ($\theta$ is a saddle), not already imply property (2) (that $\theta$ is not a local minimum)? Compare to your definition of a saddle in line 96.

**Q6:** In line 113, you refer to $ H(\theta)$, but in Eq. (4) and throughout the rest of the paper, you use $ H(x) $ instead. Could you clarify the difference between these notations and what has changed?


**Q7:** Can you specify which result in Ziyin (2023) leads to eq. (4)?

**Q8:** Proof of Thm. 1: you start by assuming eq (4), but why do you not need the projection $P$? And I thought you needed symmetry-induced saddles for this equation to hold.. are all type two saddle symmetry induced?



**Remark**

I made a genuine effort to understand the content of this paper, as I find the topic highly interesting overall. Assuming that Eq. (4) holds (without projection) around the so-called Type 2 saddles, the proofs appear solid, and the experimental results are indeed promising. I would be glad to raise my score if the questions above are addressed satisfactorily.

---

> ### Author Response · Authors · 2024-11-22
> **Rebuttal part I**
>
> **Questions:**
>
> **Q1: Sec 3. line 96: loss $l$ not introduced. I have a hard time to understand anything if I do not know the underlying problem and the probability space and SGD scheme is also not defined. Could you provide:**
> * **A formal definition of the loss function $l$.**
> * **A clear description of the probability space and problem setting.**
> * **A precise definition of the SGD update rule used**
>
> We apologize for not being clear about the problem setting up to Section 3. Our definition of SGD and the loss function is standard because it is really just the standard minibatch SGD without replacement. We have updated the manuscript in section 3 to clarify this (Eqs. 1-2).
> $\ell$ is a per-batch loss, where the sampling happens with replacement (and so there is no correlation between minibatches). The average of $\ell(\theta,x)$ over the training data distribution, $\mathbb{E}_x[\ell(\theta,x)]$ gives the empirical loss. Here, $x$ denotes a minibatch, where each data point is independently sampled. Note that the distribution of $x$ can be either a continuous distribution (in case of online SGD) or a discrete distribution (in case of a finite training set).
>
>
>
> **Q2: Eq. (1): Could you clarify the dimension of $\theta$? From the context, it seems to be a vector in $\mathbb{R}^d$, but it’s unclear how a $d$-dimensional vector could be $\mathcal{O}$ of a real value, and whether this holds for any $t$. I’m not familiar with this notation and am uncertain about its meaning here.**
>
> Yes, $\theta \in \mathbb{R}^d$ is a vector. To avoid confusion, we have modified expression (1) to $ ||\mathbb{E}[\theta_{t+1\} -\theta_t]|| = O(||\Delta \theta||)$, so it is clear that we are comparing the order of magnitude between scalars. Equation (1) is supposed to inform the reader that we denote the order of magnitude of $\theta_t - \theta^*$ by $||\Delta \theta||$. This states that the norm of the gradient is proportional to the distance.
>
>
> **Does this equation apply to any saddle point, and how is it linked to an expected gradient norm of zero? Additionally, when you refer to “$\Delta \theta$ as the Euclidean distance from the saddle,” do you mean $|\theta_t - \theta|$, or is it some quantity independent of $t$?**
>
> Yes. It applies to any saddle point. All saddle points can be described by Eq. (2) and (3), and they clearly obey this condition. And yes, $\Delta \theta : = \theta_t - \theta^*$. To clarify this point, we have changed $\Delta \theta$ to $\Delta \theta_t$.
>
>
>
> **Q3: Following Q1 and Q2, I’m having some difficulty understanding the derivation of the different saddle types. Could you provide a more rigorous derivation of the saddle types. Are there other types not covered by this classification? Could you explain why only these two types are considered, if that is indeed the case.**
>
> The definition we used is based on the linear order term in the dynamics close to a saddle. It is a simple logical consequence that there are only two types: (1) the linear order term is random and nonvanishing (type I) and (2) the linear order term vanishes with probability $1$ (type II). However, this does not mean that we cannot have more fine grained classifications. For example, one can further divide each saddle type into subtypes according to higher order terms, and this could be an interesting future direction.
>
>
> **Q4: Could you clarify what $P$ denotes the projection of— is it $\theta$ or $H$? Based on Definition 1, it seems likely that $\theta$ is intended, but this is just my assumption.**
>
> There is really no difference. Operating on $\theta$ is the same as operating on $H$. After all, projectors have the property that $P^2 =P$. Thus, $HP \theta = (HP)(P\theta)$, and so it is clear that the projection matrix is simultaneously acting on both the Hessian and parameter. That this is possible is guaranteed by Theorem 1 (Part 2) in https://arxiv.org/pdf/2309.16932. It is because when symmetry is present, the Hessian can be split to an escape direction.
>
>
>
>
> **Q5: How does Def. 1 imply eq. (3)? And does property (3) ($\theta$ is a saddle), not already imply property (2) (that $\theta$ is not a local minimum)? Compare to your definition of a saddle in line 96.**
>
> The derivation from Def. 1 to eq. (3) can be found in the answer to question 1. In the case of type-I saddle, as the gradient term does not vanish, we have $ \nabla l(x_t, \theta^*)$ in the dynamics. It is denoted as $r_t$ in equation (2). We have added a proposition in appendix B.1 to explain this.

---

> > ### Author Response · Authors · 2024-11-22
> > **Rebuttal part II**
> >
> > **Q6: In line 113, you refer to $H(\theta)$, but in Eq. (4) and throughout the rest of the paper, you use $H(x)$ instead. Could you clarify the difference between these notations and what has changed?**
> >
> > $H(\theta)$ is defined as $\nabla^2 \mathbb{E}[l(\theta,x)]$, which is a matrix-valued function of $\theta$. We have added this formula on line 110-111. Please notice that $\hat{H}(x_t)$ always comes with a hat, as it is a matrix-valued random variable depending on the data point.
> >
> > **Q7: Can you specify which result in Ziyin (2023) leads to eq. (4)?**
> >
> > The discussion in section 2.4 before equation (4) in Ziyin (2023) explains how to obtain equation (4) in our work.
> >
> > **Q8: Proof of Thm. 1: you start by assuming eq (4), but why do you not need the projection $P$? And I thought you needed symmetry-induced saddles for this equation to hold.. are all type two saddle symmetry induced?**
> >
> > We omitted the projection in equation (5) for notational simplicity, but we should not have done this. Thank you for pointing this out. We have fixed it. While symmetry is one of the primary causes of Type-II saddles, we think it is worthwhile to point out that symmetry is not the only reason that leads to a Type-II saddle and so our theory is not limited to the case where the loss function has symmetries. As an example, consider a one-dimensional two-layer GELU network $f(x) = u w x \Phi(wx)$ with l2 training loss $\frac{1}{2}(f(x) - y)^2$, the gradient for each data point is $(f(x) - y) (wx \Phi(wx), ux \Phi(wx) + uwx^2 \Phi^\prime(wx))$. $\Phi(x)$ in the expression is the cumulative distribution function of Gaussian distribution. This quantity is always zero at origin, indicating that the origin can be a type-II saddle. The origin has low rank by definition. However, there is no symmetry in this problem.
> >
> > **Remark**
> >
> > **I made a genuine effort to understand the content of this paper, as I find the topic highly interesting overall. Assuming that Eq. (4) holds (without projection) around the so-called Type 2 saddles, the proofs appear solid, and the experimental results are indeed promising. I would be glad to raise my score if the questions above are addressed satisfactorily.**
> >
> > Thanks for your honest and encouraging remark!

---

> ### Comment · Reviewer_6FcH · 2024-11-24
>
> I would like to thank the authors for their efforts in clarifying the problem settings and improving the notations in the paper.
>
> However, I still observe several instances of mathematical imprecision, leading to theoretically incorrect equations that require further attention. These inaccuracies hinder the paper’s readability and the reader’s ability to fully grasp its contributions. For this reason, I have decided not to raise my initial score, as I believe a second revision is necessary to verify the correctness of all statements and refine the presentation.
>
> Below, I provide detailed remarks and comments that I hope will serve as constructive feedback to enhance the manuscript in future revisions.
>
> **Q1:**
> While the problem statement is now defined, I still believe there is significant room for improvement in the notations to make the presentation clearer and more accessible. Specifically, introducing
>  $\ell(\theta) = \mathbb{E}_{x\sim D}[\ell(\theta,x)]$
> for the data distribution $D$ would allow you to define $H(\theta) = \nabla^2 \ell(\theta)$  formally, rather than just describing it in words. This would make the mathematical framework more rigorous and easier to follow.
>
> **Q2:** Thank you for addressing my earlier concerns. I recommend defining
> $\Delta(\theta) := ||\theta_t - \theta*||$
> and subsequently omitting the norm symbol around $\Delta(\theta) $ for notational simplicity. Additionally, it would be helpful to clarify in the main text that  $\theta*$  represents the saddle point of interest, ensuring that readers can easily follow the discussion.
>
> **Q3:** The derivation in Appendix B.1 is central to understanding the classification of saddle types. Including this derivation in the main text would significantly enhance the paper’s clarity and logical flow. The paper is much clearer to me now after reviewing this section. That said, I still encounter some issues with the equations in Appendix B.1:
> - Missing transformations or scalar products: When working with vectors, the equations should include the appropriate transformations or scalar products. And, the term at the end should be written as $O( ||\theta_t - \theta*||^2)$.
> - Inconsistency with the O-term: The O-term is omitted in Equation (14) but then reappears in Lines 984 and 989, which is confusing. Maintaining consistency is crucial for clarity.
> - Comparing to eq. (5), the O-term in line 984 should be $O( ||\theta_t - \theta*||^2)$.
>
>
> Again as a final remark: Using formally correct and precise mathematics is essential for enabling readers to follow and validate the theoretical results. I encourage the authors to address these issues carefully in the next revision.

---

> > ### Author Response · Authors · 2024-11-25
> > **Reply**
> >
> > Thanks for your constructive and very valuable feedback, and we apologize for the typos in B1. We have updated a new revision. We have taken this opportunity to incorporate B1 into section 3. We believe this revision to significantly improve the readability of Section 3 and its connection to later sections.
> >
> > **Q1, Q2**: Thanks for pointing out this problem. We have fixed it in the newly updated revision.
> >
> > **Q3**: We have fixed these problems and incorporated B1 into Section 3. This also clarifies the derivation of the original symmetry-related dynamics and improves the connection of Section 3 to later sections.
> >
> > We agree that mathematical clarity and precision are important for our work, and we are happy to follow your advice. We will also carefully check through the manuscript again before the final version to ensure everything else is mathematically precise and consistent.

---

### Official Review · Reviewer_JN3x · 2024-11-04

**Soundness:** 2
**Presentation:** 2
**Contribution:** 2
**Rating:** 5
**Confidence:** 3

**Summary:**

In this paper, the authors focus on the dynamics of stochastic gradient descent (SGD) around the saddle points. They propose a fine-grained characterization of saddle points: Type-I and Type-II, where gradient noise vanishes in Type-II saddles. The paper focuses on Type-II saddles which are intuitively harder than Type-I. By analyzing the linearized dynamics in simple settings, the authors show that there are different dynamics depending on stepsize and data. Numerical experiments are provided to support the claims.

**Strengths:**

-	Understanding the dynamics of SGD near saddle points is an interesting and important research problem
-	The classification of saddle points into Type-I and Type-II seems to be new and interesting.
-	Some of the theoretical insights from simple linearized dynamics can transfer to more complex settings also seem to be interesting.

**Weaknesses:**

Many places are not very clear to me as a reader. See questions section below.

**Questions:**

-	For eq (1), I don’t understand how to get it. Also, the definition of $\|\|\Delta\theta\|\|$ is not very clear. It is the distance from saddle point to which point?
-	In eq (3) and (4), the Hassian matrix $\hat{H}(x_t)$ is at which point? I’m assuming it is at saddle point $\theta^*$, but it does not seem to mention this in the paper.
-	For eq (4), I believe it’s worth mentioning that this is an approximate or linearized dynamic instead of the actual dynamic. The current way of writing is not clear about this.
-	In eq (5), are $\theta_t$ and $\theta_*$ scalars or vectors? The description of 1d dynamics makes me think they are scalars, but in Theorem 1 it seems they are vectors.
-	In eq (8), what are $i$ and $d$ here?
-	In Figure 2, which point is the $\Lambda$ computed at?
-	I don’t see how to get the equation in line 288. Is it assuming $\theta_*=0$? Where does the $\|\|\theta_0\|\|^2$ in the denominator come from?
-	For the experiments in section 5.1, I wonder in general how to make the predictions using $L_2$ or probability stability. This seems not discussed in the paper. As a related question, if there are multiple attractive/stable points, how should one predict which point the algorithm should converge to? For example in the description in line 317, I believe both B and C are probability stable, so why SGD converge to C instead of B?
-	I would not agree with the interpretation before Theorem 4 that all neural networks are initialized close to a Type-II saddle. The initialization is not close to the 0 point for example in the sense of Frobenius norm.
-	In Figure 6 and 7, I wonder why use rank and sparsity as the metric to measure whether SGD escapes a saddle point. It is not clear to me why these results show that ‘’Type-II saddle are indeed a major obstacle in the initial phase of training’’ (line 472).

Typo:

-	In line 43, the description of Type-I and Type-II saddle points are the same.

---

> ### Author Response · Authors · 2024-11-22
> **Rebuttal part I**
>
> **Questions:**
>
> **For eq (1), I don’t understand how to get it. Also, the definition of $||\Delta \theta||$  is not very clear. It is the distance from saddle point to which point?**
>
> The $||\Delta \theta||$ refers to the distance between $\theta_t$ and the saddle under consideration, $\theta^*$. This states that the norm of the gradient is proportional to the distance. This equation is due to a simple application of the Taylor expansion of the gradient; the leading order term of the expansion must be order $||\Delta \theta||$.
>
>
>
>
> **In eq (3) and (4), the Hassian matrix $\hat{H}(x_t)$ is at which point? I’m assuming it is at saddle point $\theta^\*$, but it does not seem to mention this in the paper.**
>
> Yes, it is at the saddle. We have mentioned it explicitly in the update.
>
>
> **For eq (4), I believe it’s worth mentioning that this is an approximate or linearized dynamic instead of the actual dynamic. The current way of writing is not clear about this.**
>
> We have mentioned it explicitly in the update.
>
>
> **In eq (5), are $\theta_t$ and $\theta^\*$ scalars or vectors? The description of 1d dynamics makes me think they are scalars, but in Theorem 1 it seems they are vectors.**
>
> The $\theta_t$ and $\theta^*$ are vectors. For 1d dynamics, we consider the case where the projection $P\theta_t$ is a scalar. We forgot the projection operator in equation (5). We have fixed this.
>
> **In eq (8), what are $i$ and $d$ here?**
>
> This is a typo. It should be $\theta_t - \theta^* = \prod_i^t Z_i (\theta_0 - \theta^*)$. We have fixed it. Thank you for pointing out this.
>
> **In Figure 2, which point is the $\Lambda$ computed at?**
>
> $\Lambda$ is computed at the origin, which is a type-II saddle.
>
>
> **I don’t see how to get the equation in line 288. Is it assuming $\theta^\* = 0$? Where does the $||\theta_0||^2$ in the denominator come from?**
>
> Yes. We assumed $\theta^* = 0$ for notational simplicity. Also, we forgot the $||\theta_0||^2$ in the denominator in the first equation. We have fixed this.
>
>
> **For the experiments in section 5.1, I wonder in general how to make the predictions using $L_2$ or probability stability. This seems not discussed in the paper. As a related question, if there are multiple attractive/stable points, how should one predict which point the algorithm should converge to? For example in the description in line 317, I believe both B and C are probability stable, so why SGD converge to C instead of B?**
>
>
> This is a very good question. This is a general limitation of stability-based analyses because the theory can only tell you where the algorithm will absolutely NOT converge to. It functions thus like a worst-case guarantee. That being said, one can have some intuitions about where the SGD might go. After all, SGD makes updates based on the local gradient information, and where SGD travels depends on the local noise. This makes it more likely for SGD to travel to the closest stable solution from its init, and so C is the more likely place to converge to, and this intuition indeed agrees with the experiment.
>
>
>
>
> **I would not agree with the interpretation before Theorem 4 that all neural networks are initialized close to a Type-II saddle. The initialization is not close to the 0 point for example in the sense of Frobenius norm.**
>
> This is a great question. The actual picture here is a little complicated. The high-level picture is that there are two competing length scales: (1) the length scale of initialization, and (2) the length scale of the origin saddle. When we say that the length scale of init is small, it is in comparison to the length scale of the origin saddle. Theoretically, it is not quite sure what is this length scale (some answer existed in, for example, https://arxiv.org/abs/2202.04777).
>
> This is some possibility of estimating this saddle length scale empirically, for example. A good rule of thumb is that if the loss function connects from the init to the origin by a monotonic change in the loss and for most of the data points, the init can be considered close to the initialization saddle. Empirically, we do find the common initialization to be within the length scale of the initialization saddle. See the new experiments in appendix A.7 to show that this is indeed the case for MLP and Resnet18. Also, the results in Figure 5, 6, 7 are also consistent with this argument.
>
> Lastly, we agree that it is certainly the case that one can find datasets for which the initialization scale is larger than the origin saddle scale, and an interesting open question is to study what influences this length scale.

---

> > ### Author Response · Authors · 2024-11-22
> > **Rebuttal part II**
> >
> > **In Figure 6 and 7, I wonder why use rank and sparsity as the metric to measure whether SGD escapes a saddle point. It is not clear to me why these results show that ‘’Type-II saddle are indeed a major obstacle in the initial phase of training’’ (line 472).**
> >
> > At origin, $\theta=0$, and so all the weights are zero rank and sparse. Thus, the sparsity and low-rankness is a quantitative metric of how close the model is to the origin.
> >
> >
> > **Typo:
> > In line 43, the description of Type-I and Type-II saddle points are the same.**
> >
> > Thank you for pointing this out. We will change it to “Type-I where the noises of the gradient do not vanish in the escape directions, and Type-II saddles where the gradient noises vanish in the escape directions.”

---

> ### Comment · Reviewer_JN3x · 2024-11-25
>
> Thank authors for providing detailed response to address my concerns. I have carefully reviewed the response as well as the comments from other reviewers. I share similar concerns with the other reviewers that certain parts of the current version remain unclear. For example, I would recommend clarifying in the paper how the theory makes predictions in Section 5.1, as explained in the authors' response. I believe make things clear will improve the paper. I will maintain my score.

---

> ### Author Response · Authors · 2024-11-25
> **Reply**
>
> Thanks for explaining your additional concern.
>
> We have updated Section 5.1 to clarify how the theoretical lines are calculated, and this should clarify any remaining ambiguity. This includes our rebuttal above of the intuition that SGD prefers closer solutions (footnote 4), and a paragraph stating how we compute the instability thresholds according to different theories for this problem (lines 353-359). We believe that this addresses the concerns you had.
>
> We are happy to hear additional criticisms of our draft.

---

### Official Review · Reviewer_nTd5 · 2024-11-06

**Soundness:** 3
**Presentation:** 3
**Contribution:** 3
**Rating:** 8
**Confidence:** 3

**Summary:**

The paper proposes a classification of saddles into two types and then focuses on the second type of saddles, which are characterized by the fact that the variance of the gradient goes to zero as one approaches the saddle. The authors study the stability properties of such saddles and show that they can actually be attractive for large learning rates, as determined by a Lyapunov exponent that can be computed empirically.

The paper then studies a few simple neural network models and observe that SGD can in certain indeed converge to such type II saddle. Interestingly, these saddles tend to have low-rank weights (because they arise as symmetry induced saddles, right?).

**Strengths:**

I really appreciate the choice to focus on type II saddles, and agree that it is common for such saddles to play an important role. I believe that many previous works on SGD have tried to avoid such saddles (for example by approximating SGD with noisy/Langevin GD or some variant thereof, thus guaranteeing that all saddles are type I and therefore easily avoidable), while type II saddles appear harder to study,  their peculiarity should not be avoided, so I appreciate this paper for going this less common direction.

The experiments are on simple models but the analysis is quite thorough with phase diagrams and comparison with the computed Lyapunov exponents.

**Weaknesses:**

The theoretical results describing conditions for stability of the saddles are in terms of the Lyapunov exponents, which I am not particularly familiar with and seem hard to work with theoretically. To be honest going into the paper I was expecting that it would be possible to characterize stability/attractivity in terms of statistics of the random gradient/Hessian at the saddle (expectations or variances) like in the solvable 1D dynamics that they present. Do the authors believe that no such general condition could be obtained? And is the section on insufficiency of norm stability trying to explain why such a condition is not possible in general?

**Questions:**

Overall, while I think that the authors did a good job explaining the intutions between the observed behavior, I find some of the definitions a bit hard to parse, and some parts unclear:
- Is the type I / II classification a strict classification in the sense that every saddle is either type I or type II?
- Why do you define Lp-norm stability to then discard it saying that it is not useful for type II saddles two sections later? I do find Theorem 2 interesting, but maybe it would make sense to define Lp-norm stability in section 4.3, rather than introducing it earlier which suggests that it is a tool that you are going to use throughout the paper. And maybe explain that Lp-norm stability is a common and useful tool to do this in general but you show that it fails for type II saddles, or something like that.
- I find the definition of Lyapunov exponent a bit unclear, when you take the max over the initialization $\theta_0$ can you take any possible initial $\theta_0$, and when you later say that the Lyapunov exponent is independent of initialization, do you mean that it is independent of $\theta_0$?

There are two papers that are quite relevant to the symmetry induced saddles and SGD and how it can lead to a low-rank/sparsity bias (https://arxiv.org/pdf/2306.04251v3 and https://arxiv.org/abs/2305.16038). Both rely on the fact that the noise of the gradient vanish along directions that would allow the network to increase the rank, thus making these symmetry induced region attractive, which is similar to the phenomenology around type II saddles. In the second paper this effect is further amplified by the presence of weight decay (as you briefly note in the solvable 1D dynamics).

---

> ### Author Response · Authors · 2024-11-22
> **Rebuttal part I**
>
> **The paper then studies a few simple neural network models and observe that SGD can in certain indeed converge to such type II saddle. Interestingly, these saddles tend to have low-rank weights (because they arise as symmetry induced saddles, right?).**
>
> Thank you for the question. We think it is worthwhile to point out that Type-II saddles are usually, but do not have to be, symmetry-induced. Consider a one-dimensional two-layer GELU network $f(x) = u w x \Phi(wx)$ with l2 training loss $\frac{1}{2}(f(x) - y)^2$, the gradient for each data point is $(f(x) - y) (wx \Phi(wx), ux \Phi(wx) + uwx^2 \Phi^\prime(wx))$. $\Phi(x)$ in the expression is the cumulative distribution function of Gaussian distribution. This quantity is always zero at origin, indicating that the origin can be a type-II saddle. The origin has low rank by definition. However, there is no symmetry in this problem.
>
> **Weaknesses:**
>
> **The theoretical results describing conditions for stability of the saddles are in terms of the Lyapunov exponents, which I am not particularly familiar with and seem hard to work with theoretically. To be honest going into the paper I was expecting that it would be possible to characterize stability/attractivity in terms of statistics of the random gradient/Hessian at the saddle (expectations or variances) like in the solvable 1D dynamics that they present. Do the authors believe that no such general condition could be obtained? And is the section on insufficiency of norm stability trying to explain why such a condition is not possible in general?**
>
> Thanks for this very important question. Yes, a main implication of the theory is that identifying the attractivity of these saddles is hard. The insufficiency part shows that distance functions cannot be used to characterize the attractivity. What remains left are two options: (1) probabilistic stability, (2) $L_p$ norm for a concave $p$ (namely, for $p<1$). What theorem 3 then shows is that characterizing the probabilistic stability is equivalent to computing the Lyapunov exponent of random matrix products, which is a well-known open problem for dimensions larger than 1. Therefore, it is difficult to compute it in general. One hope remains: that is, one might be able to characterize the attractivity with some $L_p$ norm but with $p<1$. Of course, this is not sufficient for worst cases, but it may be possible and tractable if some reasonable conditions are given. This might be an important future direction in understanding the attractivity of SGD.
>
>
>
>
>
> **Questions:**
>
> **Overall, while I think that the authors did a good job explaining the intutions between the observed behavior, I find some of the definitions a bit hard to parse, and some parts unclear:**
>
> Thanks for this criticism. We have done our best to improve the definitions and notations in the update.
>
>
> **Is the type I / II classification a strict classification in the sense that every saddle is either type I or type II?**
>
> Yes. Type-II saddle refers to saddle points with vanishing gradients for each data point. All the rest are type-I saddles. Also, we note that it is orthogonal to standard classifications between strict and nonstrict saddles. A saddle can be further classified into four types: (1) strict and type-I, (2) strict and type-II, (3) nonstrict and type-I, and (4) nonstrict and type-II. An interesting future problem is to study the interplay between these two definitions.
>
>
>
>
> **Why do you define Lp-norm stability to then discard it saying that it is not useful for type II saddles two sections later? I do find Theorem 2 interesting, but maybe it would make sense to define Lp-norm stability in section 4.3, rather than introducing it earlier which suggests that it is a tool that you are going to use throughout the paper. And maybe explain that Lp-norm stability is a common and useful tool to do this in general but you show that it fails for type II saddles, or something like that.**
>
> This is a good suggestion. We have updated the manuscript accordingly.
>
> **I find the definition of Lyapunov exponent a bit unclear, when you take the max over the initialization $\theta_0$ can you take any possible initial $\theta_0$, and when you later say that the Lyapunov exponent is independent of initialization, do you mean that it is independent of $\theta_0$?**
>
> Yes. We mean that it is independent of initialization because the max is taken over the initialization.

---

> > ### Author Response · Authors · 2024-11-22
> > **Rebuttal Part II**
> >
> > **There are two papers that are quite relevant to the symmetry induced saddles and SGD and how it can lead to a low-rank/sparsity bias (https://arxiv.org/pdf/2306.04251v3 and https://arxiv.org/abs/2305.16038). Both rely on the fact that the noise of the gradient vanish along directions that would allow the network to increase the rank, thus making these symmetry induced region attractive, which is similar to the phenomenology around type II saddles. In the second paper this effect is further amplified by the presence of weight decay (as you briefly note in the solvable 1D dynamics).**
> >
> > Thank you for introducing these papers to us. They are also papers studying the stability of SGD, and we have included them in the related works section.

---

> > > ### Comment · Reviewer_nTd5 · 2024-11-25
> > >
> > > I thank the authors for their clarifications, and for taking my suggestions into account. I am keeping my score, I think it is a good paper.

---

### Official Review · Reviewer_3mXt · 2024-11-07

**Soundness:** 2
**Presentation:** 1
**Contribution:** 2
**Rating:** 3
**Confidence:** 3

**Summary:**

The paper is focused on the important question of the role of saddles in learning dynamics. In particular, they classify saddles into two types: Type 1 and Type 2. Type-II saddles are attractive under SGD dynamics due to the so-called vanishing gradient noise. The theory and numerics are provided to support the claims.

**Strengths:**

The paper is focused on the interesting question of studying the attractivity of symmetry-induced saddles. Even though such saddles might have a negative eigenvalue in the Hessian, they suffer from degeneracy due to the symmetries.

The one-dimensional model is illustrative and explains the escape mechanism from the saddle.

Figures are done with care.

**Weaknesses:**

I think the paper needs a significant revision. I cannot read the mathematical notation as it is so I cannot evaluate the validity of the results. For example, three places where the classification of Type I vs Type II saddles are done -- all three unclear to me.

1. Line 36: what does it mean "GD noise persists at the saddle"
2. Line 43: Type-I where the **noises of the gradient vanishes in the escape directions**, and Type-II saddles where the **gradient noise vanish in the directions of escape**.

I read this sentence maybe 5 times, but the two definitions are identical up to a permutation of words.

* Can you provide a precise mathematical definition of "GD noise persists at the saddle" ?
* Can you clarify the distinction between "escape directions" and "directions of escape", or use consistent terminology if they mean the same thing ?
* Can you define these concepts explicitly when they are first introduced?


3. Line 102-103: What is $\hat{H}(x_t)$ here?

Figure 3 seems to capture the core result of the paper and hence could be put on the second page.  Can you provide a clear definition of this concept of $L_2$ stability when it is first introduced, as this seems to be an important concept in the paper ?

Why does SGD escape the Type-II saddle in Figure 1 (even though in a longer time scale). Aren't Type-II saddles are attractive under SGD by definition? Can you explicitly address this apparent contradiction and clarify the conditions under which Type-II saddles are escaped versus when they are attractive?

Finally, the attractivity of saddles and even local max is studied in Chen et al 2023 which is a very important but missing citation in this submission. They also study the symmetry-induced saddles and low-rank matrices for linear networks.

Chen, Feng, et al. "Stochastic collapse: How gradient noise attracts sgd dynamics towards simpler subnetworks." Advances in Neural Information Processing Systems 36 (2024).

How does this paper compare to Chen et al 2023?

**Questions:**

See above.

---

> ### Author Response · Authors · 2024-11-22
> **Rebuttal part I**
>
> **Weaknesses:**
>
> **I think the paper needs a significant revision. I cannot read the mathematical notation as it is so I cannot evaluate the validity of the results. For example, three places where the classification of Type I vs Type II saddles are done -- all three unclear to me.**
>
> Thanks for this question. We have clarified these points below and in the updated manuscript. Definition 1 has no ambiguity, and we believe it is clear. We also clarified the notations above definition 1, and this should be clear now. There are some typos in the introduction, and they have now been fixed; this should remove all confusion.
>
> Also, we believe that the proofs are both rigorous and solid. This point has been acknowledged by Reviewer 6FcH, who states that assuming that the derivations up to Eq (4) are correct, the rest of the theory is solid and in promising agreement with the experiments. With our updated clarification of the notations and definitions up to Eq (4), we believe this problem should be resolved.
>
> **Line 43: Type-I where the noises of the gradient vanishes in the escape directions, and Type-II saddles where the gradient noise vanish in the directions of escape. I read this sentence maybe 5 times, but the two definitions are identical up to a permutation of words.**
>
> Thanks for this criticism. There is an unfortunate typo in the second sentence: the first part should be "Type-I where the noises of the gradient **do not** vanish in the escape directions." We have fixed this sentence.
>
>
>
> **Line 36: what does it mean "GD noise persists at the saddle." Can you provide a precise mathematical definition of "GD noise persists at the saddle" ?**
>
>
> Let $\theta*$ be a saddle. Let $P$ be an orthogonal projection whose image is the escaping directions from the saddle. That "GD noise persists at the saddle" means that there exists a data point $x$ such that $P\nabla_\theta \ell(\theta, x) \neq 0$. We have clarified the definitions in the manuscript (footnote 1).
>
>
>
>
> **Can you clarify the distinction between "escape directions" and "directions of escape", or use consistent terminology if they mean the same thing ?**
>
> These two are the same things. The confusion comes from what we explained above: there is a typo in the first part of the sentence. We have fixed this in the update.
>
>
> **Can you define these concepts explicitly when they are first introduced?**
>
> We will refer to def 1 when they are first introduced. However, we prefer to keep the formal definition out of the introduction. With the clarified phrasing, the reader should be able to build some intuition from our description.
>
> **Line 102-103: What is $\hat{H}(x_t)$ here?**
>
> $\hat{H}(x_t) = \nabla l(\theta^*, x_t)$ is the Hessian matrix at the saddle with the data point sampled at timestep $t$. Therefore, for different samplings of minibatch, $H$ will be different. The hat notation implies that $H$ can be seen as an estimate of the expected Hessian.
>
>
>
> **Figure 3 seems to capture the core result of the paper and hence could be put on the second page. Can you provide a clear definition of this concept of  stability when it is first introduced, as this seems to be an important concept in the paper ?**
>
>
> Thank you for the proposal. While we would like to put Figure 3 up in the front, it is better not to do so for the readability of the paper. Figure 3 is difficult to understand without first introducing the Lyapunov exponent, which only appeared on page 5. The notion of probabilistic stability is formally defined in Section 3. While we did our best to introduce it as early as possible, it is impossible to move it even earlier because it will not become relevant before we introduce the Type-II saddles.
>
>
>
>
>
> **Why does SGD escape the Type-II saddle in Figure 1 (even though in a longer time scale). Aren't Type-II saddles are attractive under SGD by definition? Can you explicitly address this apparent contradiction and clarify the conditions under which Type-II saddles are escaped versus when they are attractive?**
>
> Thanks for this important question. What we showed is that the Type-II saddles are attractive at a high gradient noise (namely, a high learning rate or small batch size). For example, there is a critical learning above which a Type-II saddle becomes attractive. This critical learning rate is given in Theorem 1. Thus, Type-II saddles can be either attractive or repulsive (in contract, Type-I can only be repulsive).

---

> > ### Author Response · Authors · 2024-11-22
> > **Rebuttal part II**
> >
> > **Finally, the attractivity of saddles and even local max is studied in Chen et al 2023 which is a very important but missing citation in this submission. They also study the symmetry-induced saddles and low-rank matrices for linear networks. Chen, Feng, et al. "Stochastic collapse: How gradient noise attracts sgd dynamics towards simpler subnetworks." Advances in Neural Information Processing Systems 36 (2024). How does this paper compare to Chen et al 2023?**
> >
> > Chen et al. 2023 have some overlap with our work. There are two major differences: the analysis of Chen et al. 2023 is based on the stochastic differential equation approximation, while we directly tackle discrete-time SGD -- which is what is actually used in practice. We have included Chen et al. 2023 in the related works section. Also, the main insights and focus are different. We focus on the classification of saddles, which is not present in Chen et al. (2023).

---

> > > ### Comment · Reviewer_3mXt · 2024-11-26
> > > **thanks**
> > >
> > > I thank the authors for their response. Def 1 makes clear now the distinction between Type-I and Type-II saddles --- methods based on approximating SGD with SDE would fail to make this classification indeed.
> > >
> > > I still do not understand why SGD escapes the Type-II saddle in Figure 1. Is it because a small learning rate or a large batch is used here?
> > >
> > > I've read Proposition 1 which is better than before but not yet formal in my opinion. The text reads, at a Type-I saddle, then Eq (6) but the saddle $\theta^*$ does not appear in Eq 6. This Proposition needs significant improvement and the manuscript. In the proof, there comes the definition of Type-I saddle once again, for example.
> > >
> > > Another point is Eq (12) is the same as Eq (5) ?! It may be clear to the authors for some very mysterious reasons but it looks like a random claim to the reader.
> > >
> > > I think there are some nice contributions in the paper, that is why I made another pass. However, the updated text is still very raw, and the paper needs a major update in writing math.

---

> ### Author Response · Authors · 2024-11-28
> **Thanks and reply**
>
> Thanks for your additional reply and thoughtful comment. Some of the intended changes slipped through our first revision, and we apologize for this. We have updated the manuscript again (especially Section 3) to ensure precision and clarity. A few other terminologies and notations are also being explained in a little more detail in this update. Also, we would like to point out that while part of your criticism is valid (about definition 1), another part of it is due to misunderstanding.
>
>
> **I still do not understand why SGD escapes the Type-II saddle in Figure 1. Is it because a small learning rate or a large batch is used here?**
>
> Thanks for this question. This is because a small learning rate is used. Operationally, this is not difficult to achieve because the learning rate just has to be sufficiently small (so if it does not escape, we just use an even smaller learning rate), as justified by the argument below the last equation in Section 4. After all, the purpose of this figure is to show the generic difficulty in escaping type-II saddles in comparison to type-I. This is clearly what the figure demonstrates. Regardless of whether it escapes or not, the message is unchanged (if it does not escape, the figure shows even greater difficulty).
>
>
>
>
>
> **I've read Proposition 1 which is better than before but not yet formal in my opinion. The text reads, at a Type-I saddle, then Eq (6) but the saddle $\theta^\*$ does not appear in Eq 6.**
>
> We are afraid to say that this criticism is false. $\theta^*$ does appear in Eq 6. Simply note that the term $r_t$ is defined as $r_t = P\nabla_\theta \ell (\theta^* , x_t)$ below Eq 6.
>
>
> **This Proposition needs significant improvement and the manuscript. In the proof, there comes the definition of Type-I saddle once again, for example.**
>
> Thanks for this criticism, and we apologize for not getting this point fixed in our first revision. We were planning to define Type-I saddle in definition 1 in our revision, but this slipped through our revision. We have now fully revised Definition 1 and Proposition 1 to ensure precision and clarify. Now, Type-I saddle is not defined in the proof but only invoked to prove things (which should have been the case).
>
>
> **Another point is Eq (12) is the same as Eq (5) ?! It may be clear to the authors for some very mysterious reasons but it looks like a random claim to the reader.**
>
> Thanks for this question. This follows from simple arithmetic that can be proved in just one line. For potentially confused readers, we have included its derivation in Eq. (12).

---

### Author Response · Authors · 2024-11-22
**General reply**

We would like to thank all the reviewers for their valuable feedback. We are glad to see that the reviewers are recognizing our contributions as important and novel, specifically the classification of saddle points and understanding the dynamics of SGD near saddle points. We believe this classification will benefit both the broad field of stochastic optimization in general and the field of deep learning theory. Furthermore, this novel classification bridges what is known in conventional control theory and ergodic theory to the study of the optimization of neural networks.

In the meantime, we acknowledge that our original manuscript had flaws. We thank all the reviewers for their questions and suggestions. We have revised the paper to address the concerns we received, with a particular focus on improving the readability of the manuscript. Regarding the frequently identified problems and asked questions:

1. We have clarified the notations, definitions, and problem setting in Section 3 - while the original derivation up to Eq (4) has been a little confusing, we believe the rest of the theory is entirely solid (as also pointed out by Reviewer 6FcH)
2. We have performed a new experiment to show that common initialization schemes indeed place models close to a Type-II saddles

The reviewers pointed out various typos, which we have fixed. We have also included the references pointed out by the referees to the manuscript, specifying briefly the similarities and differences between their work and ours. Also, note that we have introduced the new Equations 1 and 2, and so the equation numbers are shifted from the original version by 2. Below, we address the questions of each reviewer in detail.

We believe that the manuscript is now significantly improved, and we encourage the reviewers to ask additional questions, and we are happy to address them.


-----------

--------
**Update**: Following an additional advice from 6FcH, we have rewritten section 3 to be more precise about our problem setting. This involves the new Proposition 1, which formally derives the starting equations that our theory relies on.

**Update 2**: Following an additional advice from JN3x, we have update section 5.1 to clarify how the theoretical predictions are calculated based on our and previous results. A limitation of our theory is also included in footnote 4. This should greatly improve the precision of our results

**Update 3**: Following an additional criticism from 3mXt, we have revised section 3, especially Definition 1 and Proposition 1, to improve their precision and clarify. This part of the revision was intended in update 2 but unfortunately slipped our attention. We apologize for this missing

Lastly, we would like to thank all the reviewers again for their careful and engaging discussions with us. We really appreciate your effort in improving our manuscript.

---

### Meta-Review · Area_Chair_Auon · 2024-12-19

**Metareview:**

The paper investigates the dynamics of SGD around saddle points, distinguishing two types of saddles and introducing a novel framework using random matrix theory and Lyapunov exponents. While the theoretical insights into the behavior of SGD near saddle points are interesting, reviewers raised significant concerns about the clarity of some theoretical arguments. Despite the authors' rebuttal, these concerns remained unresolved. As a result, I recommend rejection.

**Additional Comments On Reviewer Discussion:**

During the discussion, reviewers highlighted concerns regarding the clarity of the writing and the rigor of the theoretical analysis, which made the contributions difficult to assess. While the authors attempted to address these points in the rebuttal, their responses were found insufficient to fully resolve the reviewers’ concerns. The lack of precision in key arguments and presentation issues ultimately outweighed the strengths of the proposed framework. These points were central to my final decision to recommend rejection.

---

### Decision · Program_Chairs · 2025-01-22

Reject